# The dual role of amyloid-β-sheet sequences in the cell surface properties of *FLO11*-encoded flocculins in *Saccharomyces cerevisiae*

Clara Bouyx[1], Marion Schiavone[1,2], Marie-Ange Teste[1], Etienne Dague[3], Nathalie Sieczkowski[2], Anne Julien[2], Jean Marie François[1]*

[1]Toulouse Biotechnology Institute, INSA, Toulouse, France; [2]Lallemand, Lallemand SAS, Blagnac, France; [3]LAAS, CNRS, Toulouse, France

**Abstract** Fungal adhesins (Als) or flocculins are family of cell surface proteins that mediate adhesion to diverse biotic and abiotic surfaces. A striking characteristic of Als proteins originally identified in the pathogenic *Candida albicans* is to form functional amyloids that mediate *cis*-interaction leading to the formation of adhesin nanodomains and *trans*-interaction between amyloid sequences of opposing cells. In this report, we show that flocculins encoded by *FLO11* in *Saccharomyces cerevisiae* behave like adhesins in *C. albicans*. To do so, we show that the formation of nanodomains under an external physical force requires a threshold number of amyloid-forming sequences in the Flo11 protein. Then, using a genome editing approach, we constructed strains expressing variants of the Flo11 protein under the endogenous *FLO11* promoter, leading to the demonstration that the loss of amyloid-forming sequences strongly reduces cell-cell interaction but has no effect on either plastic adherence or invasive growth in agar, both phenotypes being dependent on the N- and C-terminal ends of Flo11p. Finally, we show that the location of Flo11 is not altered either by the absence of amyloid-forming sequences or by the removal of the N- or C-terminus of the protein.

*For correspondence:
fran_jm@insa-toulouse.fr

Competing interest: The authors declare that no competing interests exist.

## Introduction

The yeast cell wall is a highly dynamic structure that is not only an armour separating cell from its surrounding but that is endowed of surface properties including adherence to inert material leading to biofilm formation, cell-cell adhesion that can yield to flocculation, hydrophobicity which may result in buoyant biofilms. These properties are mediated by a variety of surface proteins called adhesins (*Lipke et al., 2018*), among which flocculins are adhesins encoded by the *FLO* genes family (*FLO1, FLO5, FLO9, FLO10,* and *FLO11*) in the yeast *Saccharomyces cerevisiae* (*Verstrepen and Klis, 2006*; *Brückner and Mösch, 2012*). In the pathogenic yeast *Candida albicans*, it has been reported that these adhesins can be organized into clusters of hundreds of proteins at the cell surface leading to adhesion nanodomains. This 3D organization of adhesins into nanodomains has been shown to be triggered by an external physical shear force, such as the extension force of single molecules stretched by an atomic force microscopy (AFM) tip, which then propagates across the cell surface at a speed of about 20 nm/min (*Alsteens et al., 2010*). This unique phenomenon was explained by the presence of small amyloid-core sequences of five to seven residues (IVIVATT) in adhesin proteins that are also characterized by a high content of β-branched aliphatic Ile, Val, and Thr forming β-aggregates structures as predicted by the β-aggregate predictor TANGO (*Fernandez-Escamilla et al., 2004b*; *Lipke et al., 2018*). The formation of these nanodomains is therefore an emergent property of adhesin primary sequence that promotes cell-cell aggregation, resulting in the formation of

robust biofilms and eventually promotes fungal infection (*Lipke et al., 2018*). More recently, Lipke, Dufrêne, and colleagues developed a new pipette technology called fluidic force microscopy enabling to measure forces between single cells (*Dehullu et al., 2019b*). Using this methodology, these authors showed that amyloid-forming sequences have a dual role, namely in the formation of adhesin clusters (nanodomains), which correspond to *cis*-interactions between molecule of the same cell, and in homophilic adhesion between adhesins on opposing cells (*trans*-interaction) (*Ho et al., 2019*; *Dehullu et al., 2019a*).

In a previous work aiming to investigate the impact of autolysis process on the nanomechanical properties of the cell wall of different *S. cerevisiae* strains, we identified by AFM the presence of abundant patches with a mean diameter of 140 nm at the surface of an industrial wine yeast strain (*Schiavone et al., 2015*). In addition, we found that these patches were formed from highly mannosylated proteins as they interacted strongly with a concanavalin A functionalized AFM tip, showing distances rupture ranging from 50 to 500 nm. Overall, these nanostructures were reminiscent of the adhesion nanodomains formed by the clustering of Als5 adhesins on the surface of *C. albicans* (*Alsteens et al., 2010*). In this work, we addressed the question of whether the formation of these patches could result from the aggregation of flocculins, and if this is the case, then which of the flocculins could be responsible, taking into account that a transcriptomic analysis carried out in L69 strain revealed notably a high expression of *FLO11*, although *FLO10*, *FLO5*, and *FLO1* were also expressed albeit at lower levels (*Schiavone et al., 2015*). These flocculins share a common three-domain structure. The C-terminal part (C-domain) contains a glycosylphosphatidylinositol (GPI) anchoring site for covalent attachment of the protein to β (1,6)-glucans of the inner cell wall network (*Lu et al., 1995*; *Bony et al., 1997*; *Kapteyn et al., 1999*). The middle part, stalk-like B-domain, contains serine and threonine enriched tandem repeats (TRs) encoded by conserved DNA sequences that serve as a source of variability triggered by frequent recombination events by which new adhesion/flocculin alleles could be generated (*Verstrepen et al., 2005*). The N-terminal part (A-domain) which protrudes from the cell surface allows to classify these flocculins into two groups based on their phenotypic properties. The first type includes Flo1, Flo5, Flo9, and Flo10 proteins, which mediates flocs formation. This phenotype is brought about by the A-domain that harbours a unique $Ca^{2+}$ binding motif DcisD for carbohydrate binding (*Kraushaar et al., 2015*; *Veelders et al., 2010*). The other group is represented only by Flo11p, which is implicated in several phenotypes including filamentation, flor and biofilm formation (reviewed in *Brückner and Mösch, 2012*). Like *C. albicans* adhesins, these *S. cerevisiae* flocculins contain β-aggregation sequences of five to seven amino acids length that are rich in Ile, Val, and Thr. However, the distribution of these repeats along the flocculins sequence enables to distinguish Flo1p, Flo5p, and Flo9p from Flo10p and Flo11p. For the three former proteins, the β-aggregation-positive sequences predicted by TANGO software (http://tango.crg.es/) revealed a large number of 'T(V/I)IVI' motifs that are widely distributed over the B-domain (see details in *Supplementary file 1a*). Interestingly, a synthetic peptide 'TDE<u>TVI</u>-<u>VI</u>RTP' containing this motif was shown to form amyloid fibers in vitro (*Ramsook et al., 2010*). On the other hand, β-aggregation-prone sequences are less abundant in Flo10p and Flo11p and they are all located in the C-terminus. Nevertheless, it was reported that a 1331-residue soluble Flo11p can assemble into amyloid fibers in vitro and this feature was ascribed to the presence of the amyloid-β-aggregation-prone sequences 'VVSTTV' and 'VTTAVTT' at the C-terminus of Flo11p (*Ramsook et al., 2010*).

The purpose of this work was therefore to investigate the molecular basis and the physiological function of these patches formed on the cell surface of this industrial wine yeast strain. This study was furthermore motivated by the fact that such abundant nanostructures have never been physically observed before in *Saccharomyces* species, despite the finding that cell-cell aggregation are potentiated under hydrodynamic shear forces and these interactions are antagonized by anti-amyloid dyes (*Ramsook et al., 2010*; *Chan and Lipke, 2014*; *Chan et al., 2016*). Collectively, the reported results lend support to the model that amyloid-forming sequences play a dual role in Flo11p-mediated cell-cell adhesion, namely promoting *cis*-interaction between Flo11p molecules in the same cell, leading to adhesive nanodomains, and contributing to *trans*-interaction between opposing cells. They further showed that this dual role is highly dependent on the number of amyloid-core sequences present in the Flo11 protein.

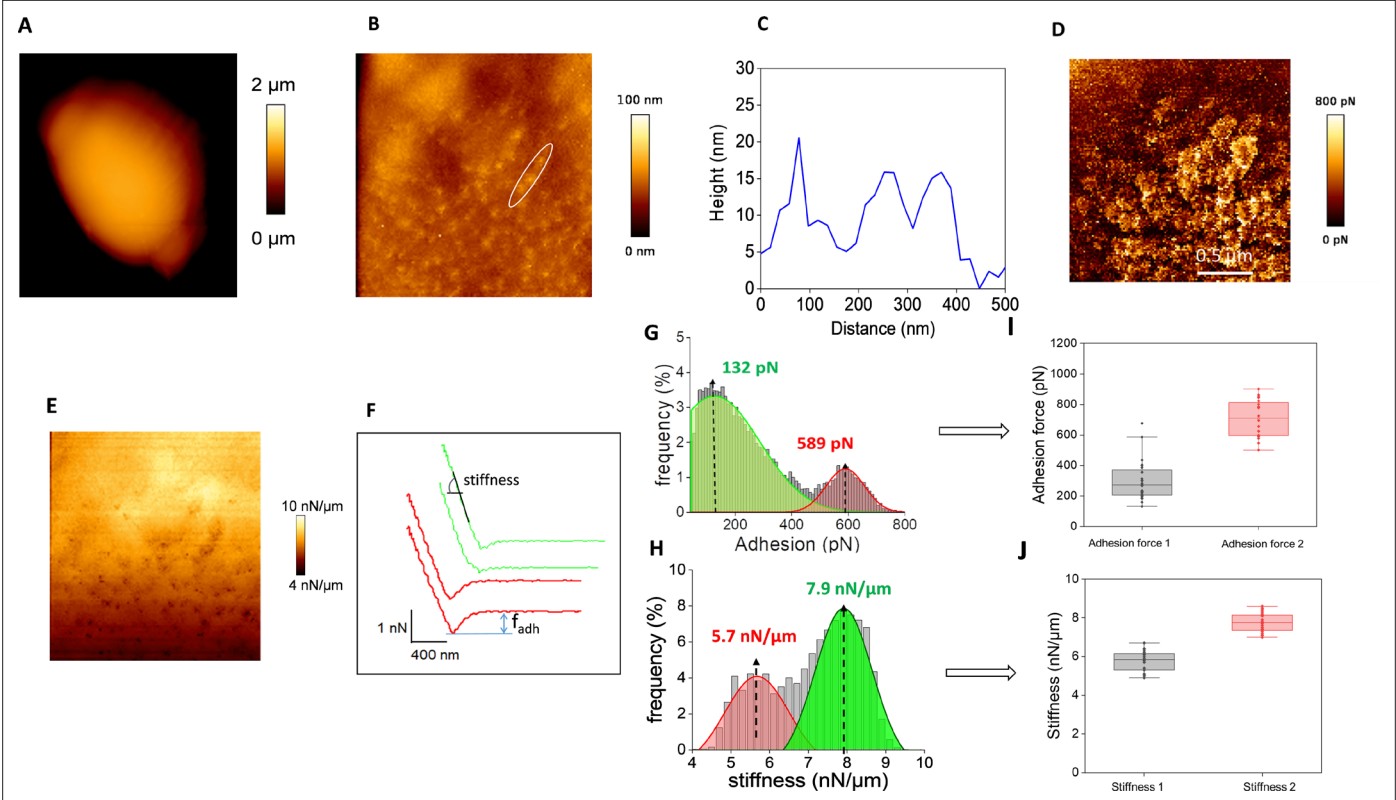

**Figure 1.** Cell surface analysis of the L69 strain using silicon nitride (Si3N4) atomic force microscopy (AFM) cantilevers. AFM height images (**A**) of a single yeast cell from L69 strain embedded in polydimethylsiloxane (PDMS) microtimber are shown. In (**B**) an AFM contact image is shown in which the topography region that was used to measure the height of the patches at the cell surface reported in (**C**) is zoomed in with a white circle. . In (**D**) is illustrated an adhesion image at high resolution of the zoomed area shown in (**C**) and the stiffness on this zoomed area is shown in **E**. (**F**) Shows a typical force-distance curve from which adhesion force stiffness was determined. In (**G** and **H**) are reported the distribution of adhesion forces (**G**) and stiffness (**H**) as obtained from 1024 force-distance curves on the single cell shown in this figure. In (I and **J**) are reported the boxplots of maximum adhesion forces and stiffness values from bimodal distribution collected from n = 416,666 force-distance curves on 24 cells from three independent experiments. The boxplots represented the mean values (squares), the medians (horizontal lines), the 25% and 75% quartiles (box limits), and outliers (whiskers).

The online version of this article includes the following figure supplement(s) for figure 1:

**Figure supplement 1.** Cell surface analysis of BY4741 and L69 strains using silicon nitride (Si3N4) atomic force microscopy (AFM) cantilevers.

## Results

### Biophysical characterization of the nanostructures at the cell surface of L69 strain

The presence of abundant patches at the cell surface of a wine yeast L69 strain under the contact of an AFM bare tip described in a previous work (*Schiavone et al., 2015*), which were totally absent on the surface of the laboratory strain BY4741 (*Figure 1—figure supplement 1*), prompted us to examine in detail the physical properties of these nanosized patches using AFM in quantitative imaging (QI) mode as this technique enables to image at high resolution and to quantify adhesive properties of the cell surface (*Chopinet et al., 2013*). *Figure 1* reports height and adhesion images of a single cell from this L69 strain trapped in the polydimethylsiloxane (PDMS) microchamber. The high-resolution height image (*Figure 1B*) revealed a multitude of small aggregates on the cell surface with an average diameter of 100 nm and a height above the cell surface in the range of 15–20 nm (*Figure 1C*). These dots are even more visible on the adhesion image at high resolution (*Figure 1D*), whereas image of *Figure 1G* illustrates the stiffness of the cell. A total of 1024 force-distance curves were collected from the surface of this cell to quantify the nanomechanical properties, namely adhesion forces which correspond to the retraction of the AFM tip from the surface and stiffness as the slope of the linear portion of the force versus indentation curve (*Figure 1F*). These two physical parameters were reported as a function of the frequency of the interaction of the tip on the surface. Data of this analysis reported

in *Figure 1G and H* showed a bimodal distribution of adhesion forces and stiffness, indicating the existence of two types of interactions. This bimodal distribution was a general feature of all cells since we collected 1024 force-distance curve for an additional 24 cells from three independent experiments, leading to similar bimodal distribution. The maximal values of adhesion forces and stiffness calculated from each cell were reported as boxplots, confirming the occurrence of two components in these patches (*Figure 1I & J*). The first interacting component that exhibits a high adhesion force (750 ± 150 pN) and low stiffness likely corresponds to hydrophobic interactions (*Dague et al., 2007*), whereas the other component characterized by a weaker adhesion forces and higher stiffness could be attributed to cell surface proteins that unfold upon retraction of the AFM tip. In summary, this AFM analysis showed that the nanoscale spots on the cell surface of strain L69 exhibit nanomechanical characteristics very similar to those of adhesin nanodomains in *C. albicans* (*Alsteens et al., 2010*; *Formosa et al., 2015b*), and raised the question of whether a same cell wall component is responsible for formation of these patches, as it is the case in *C. albicans*.

## The formation of nanodomains are inhibited by anti-amyloid agents and are totally abolished upon deletion of *FLO11*

The model of *Lipke et al., 2018*; *Lipke et al., 2012* argues that the formation of adhesion nanodomains requires amyloid-β-aggregation-prone sequences in the amino acids sequence of the involved proteins and treatment of cells with anti-amyloid agents such as a synthetic anti-amyloid peptide or amyloidophilic dyes like thioflavine S were shown to disrupt in vivo formation of nanodomains (*Alsteens et al., 2010*; *Lipke et al., 2012*; *Ramsook et al., 2010*; *Chan et al., 2016*). Accordingly, a treatment of the yeast cell with the disrupting peptide 'VASTTVT', which is the mutated motif of

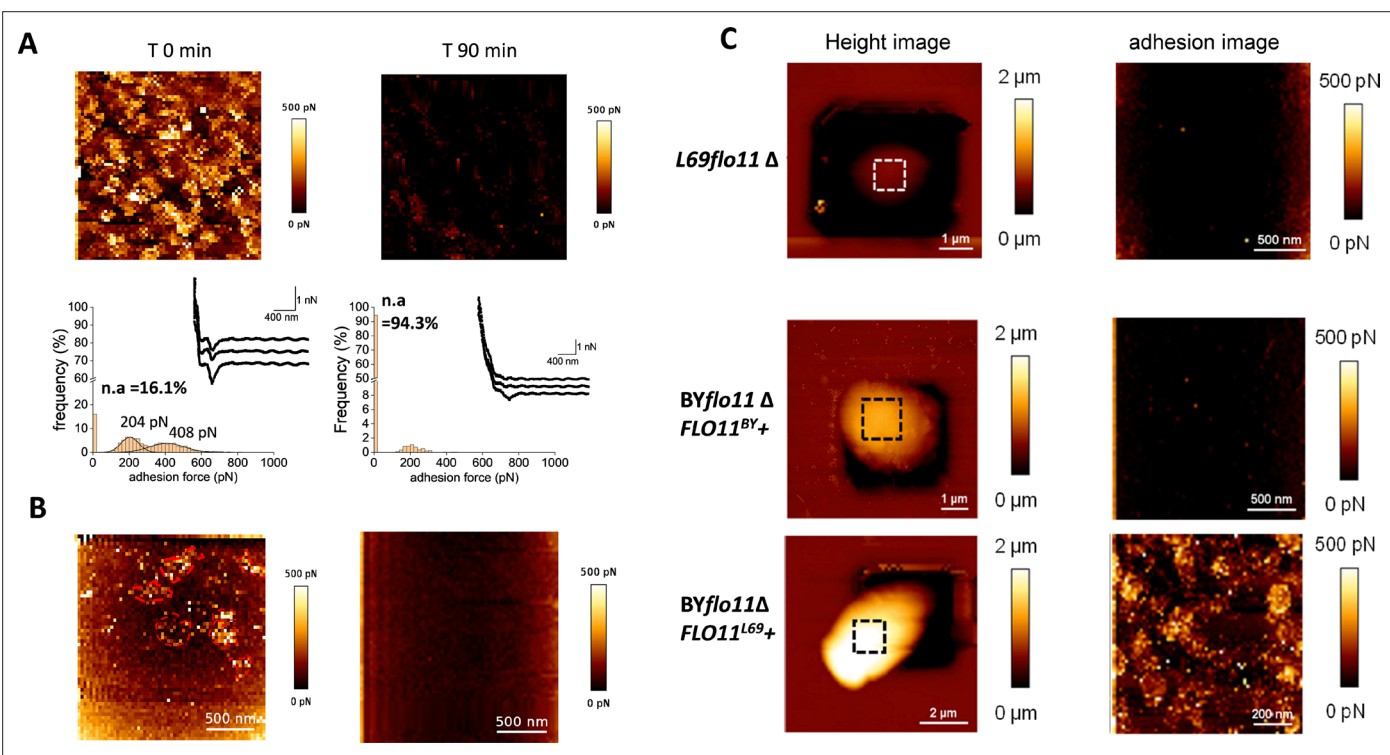

**Figure 2.** The nanosized patches at the cell surface of L69 strain are *FLO11*-dependent adhesion nanodomains that are abolished by anti-amyloïd compounds. In (**A**) is shown the atomic force microscopy (AFM) adhesion image of a cell from L69 strain before and 90 min after treatment with 5 µM of the anti-amyloid peptide VASTTV. In (**B**) is shown the adhesion image of a cell from a culture of L69 strain (10⁷ cells/ml) before and after 30 min of incubation with 10 µM of the anti-amyloid dye thioflavin S. In (**C**) are AFM height images and adhesion images of a single cell of the *flo11Δ* mutant from L69 strain and of the laboratory BY4741 strain deleted of its endogenous *FLO11* gene (BY*flo11Δ*) and transformed with pYES 2.1 carrying *FLO11* of (BY*flo11Δ FLO11^BY*+), or *FLO11* from L69 (BY*flo11Δ FLO11^L69*+).

The online version of this article includes the following figure supplement(s) for figure 2:

**Figure supplement 1.** Expression levels of *FLO11* in L69 and BY4741 strain measured by quantitative reverse transcription PCR (qRT-PCR).

the native 'VVSTTV' of Flo11p from BY4741 strain (**Ramsook et al., 2010**), for 90 min resulted in a complete disappearance of the high adhesion component of the nanodomains, while the low adhesion component was still present although reduced by about 75 % (**Figure 2A**). On the other hand, a 30 min treatment with 10 µM of the anti-amyloid agent thioflavine S abolished both components of these patches (**Figure 2B**). Taken together, these data support the idea that the patches on the surface of L69 cells are nanodomains formed by two components, one is weakly adhesive and independent of amyloid-forming sequences, and the other is highly adhesive and dependent on amyloid sequences. This result is similar to what has been reported in the case of nanodomains produced by adhesins in *C. albicans* (**Formosa et al., 2015b**).

Based on previous work showing that *FLO11* was the most highly expressed gene of the *FLO* family in strain L69 (**Schiavone et al., 2015**), and with the fact that the Flo11p of BY4741 (a derivative of S288C) is not expressed due to a non-sense mutation in the main transcriptional regulator encoded by *FLO8* (**Liu et al., 1996**; **Kobayashi et al., 1999**), we asked whether the formation of these nanodomains was due to the expression *FLO11* in L69 strain. To this end, the two copies of *FLO11* was deleted using the CRISPR-Cas9 toolbox (**Mans et al., 2015**) since L69 strain is diploid (data not shown), and this genomic deletion was confirmed by quantitative reverse transcription PCR (qRT-PCR) since L69*flo11Δ* expressed any longer *FLO11* transcript (see **Figure 2—figure supplement 1**). As reported in **Figure 2C**, loss of *FLO11* function completely abrogated the formation of nanodomains on the surface under the action of the AFM tip. This result also indicated that the two physical components that characterized these nanodomains are mainly due to the expression of *FLO11*. These data could also explain the lack of nanodomains formation in the laboratory BY4741 strain (see **Figure 1—figure supplement 1**) since *FLO11* is not expressed in this strain. To find out whether the absence of Flo11p explains the lack of nanodomains formation in the BY4741 strain, the *FLO11* gene of this strain was cloned into the high copy pYES2.1 plasmid under the dependence of the *GAL1* promoter. However, contrary to expectation, the expression of *FLO11* did not lead to the formation of nanodomains under the action of the AFM tip (**Figure 2C**), although this gene was highly expressed as determined by qRT-PCR (see **Figure 2—figure supplement 1**). This failure could be ascribed either to the lack of translation of the *FLO11* transcript or more likely to the fact that FLo11p in strain L69 differed from that of strain BY4741. To validate this suggestion, the *FLO11* gene from strain L69 whose genome was recently sequenced (Lallemand Inc, unpublished data) was retrieved in order to be expressed in BY strain from a 2µ copy plasmid as above. Interestingly, we noticed that the sequence of *FLO11* gene (provisional GenBank accession number BankIt2416107 Seq1MW448340) of strain L69 was 1.06 kbp longer than that of BY4741 (i.e. 5.16 vs. 4.10 kbp). In addition, we found that BY*flo11Δ* cells (BY4741 deleted for *FLO11* by replacement with the KanMX4 cassette) transformed with pYES2.1 carrying *FLO11^L69^* were able to form nanodomains under the action of the AFM tips (**Figure 3C**). These results taken together indicated that the formation of nanodomains does not solely depend on the expression of *FLO11* but that the encoded protein must display some specific features that are present in Flo11p of strain L69 and not in strain BY4741.

## The Flo11p of strain L69 differs from other Flo11p by a repeat amino acids sequences near the C-terminus that brings additional amyloid-forming sequences

Using Clustal Omega, we aligned the amino acids sequence of Flo11p from L69 strain (provisional GenBank accession number BankIt2416107 Seq1MW448340) with that of genome-sequenced strain S288c (**Goffeau, 1998**), Σ1278b (**Dowell et al., 2010**), a strain widely studied for its remarkable properties of colony morphology, invasive and pseudohyphal growth (**Dowell et al., 2010**; **Reynolds and Fink, 2001**; **Voordeckers et al., 2015**) as well as with the flor strain 133d reported to form buoyancy biofilms at the top of the culture medium (**Fidalgo et al., 2006**). We found that the amino acids sequence of Flo11p^L69^ is 355 and 512 longer than that of Flo11p^BY^ and Flo11p^Σ^, respectively, but only 92 longer than Flo11p from strain 133d (**Figure 3** and see **Figure 3—figure supplement 1** illustrating the sequence alignment of these Flo11p). This comparative alignment also showed that Flo11p^L69^ has an additional sequence of about 200 amino acids near the C-terminus that is completely absent in the other Flo11 proteins (see **Figure 3—figure supplement 1**, sequence highlighted in red in sequence of *FLO11^L69^*). Furthermore, a BLAST analysis performed on all *S. cerevisiae* strains sequenced to date did not reveal the presence of this additional amino acids sequence in Flo11p of these sequenced strains.

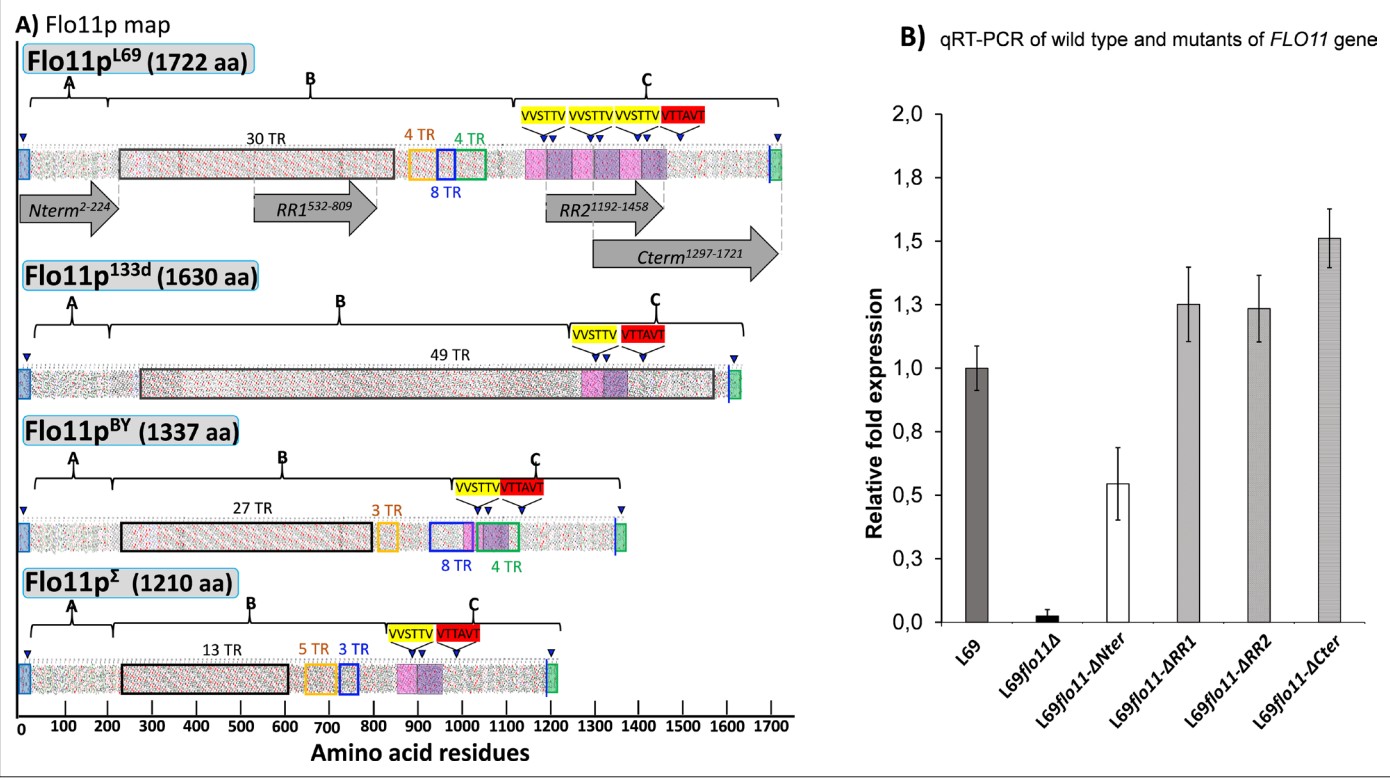

**Figure 3.** Hydrophobic cluster analysis (HCA) of the Flo11 protein from various *Saccharomyces cerevisiae* strain, and quantitative reverse transcription PCR (qRT-PCR) of wild-type and mutated *FLO11* genes. In (**A**) is reported the HCA plots of Flo11p from wine yeast strain L69, Flor strain 133d, and laboratory strains BY4741 and Σ1278b. The three domains of the proteins as determined using various software as described in Materials and methods are highlighted by letters A, B, and C. Tandem repeat (TR) domains are shown as unshaded boxes with each colour addressed to a specific repeat. Blue triangles indicate regions with a β-aggregation potential superior to 30 % in TANGO software and amyloid-forming sequences are indicated in yellow for 'VVSTTV' and red for 'VTTAVT'. Pink and purple boxes stand for two sequence repeated three times in Flo11p[L69] and present only one time in the Flo11p of the other three strains. The C-terminal glycosylphosphatidylinositol (GPI) signal is boxed in green with a blue line indicating the omega-site position (GPI signal anchorage to cell wall β-glucan). Grey arrows delimit the N-terminal, RR1, RR2, and C-terminal domains. In (**B**) is reported the quantitative expression levels of the different *FLO11* alleles encoding the corresponding Flo11 protein variants relative to the expression level of wild-type *FLO11* in L69 strain. Samples for this qRT-PCR were taken in exponential growth phase on YN galactose medium. Normalization of transcripts was done using *TAF10* and *UBC6* as internal reference as described in Materials and methods.

The online version of this article includes the following figure supplement(s) for figure 3:

**Figure supplement 1.** Sequences alignment of Flo11p from L69, BY4741, S1278b, and 133d strains.

**Figure supplement 2.** Schematic representation of the Flo11 protein sequence from BY4741 strain by hydrophobic cluster analysis.

**Figure supplement 3.** Schematic representation of the Flo11 protein sequence from S1278b strain by hydrophobic cluster analysis.

**Figure supplement 4.** Schematic representation of the Flo11 protein sequence from L69 strain by hydrophobic cluster analysis (HCA).

**Figure supplement 5.** Schematic representation of the Flo11 protein sequence from flor yeast strain 133d by hydrophobic cluster analysis.

We then carried out a hydrophobicity cluster analysis (HCA) tool as this tool is able to make emphasis on similarity and disparity of proteins having repeated sequences and high percentage of hydrophobic amino acids (*Dranginis et al., 2007*). As expected, the three-domain structure of Flo11 protein of these different yeast strains could be easily recognized from these HCA and thus schematically drawn as reported in *Figure 3* (see *Figure 3—figure supplements 2–5* for details). However, this analysis revealed major differences between the Flo11p of strain L69 and that of other yeast strains. In particular, Flo11p of strain L69 contains more repeats (TRs) than the proteins of BY4741 and 1278b, but slightly fewer (46 vs. 49) than the Flo11p of flor strain 133d. Quite interestingly, the 49 TRs in Flo11p of this flor strain are composed of the same 81 nucleotides (nt), as indicated by sequence analysis using EMBOSS TANDEM software (*Durbin et al., 2000*), whereas the other Flo11p have four types of intragenic repeats of different nt size that score differently each (see *Supplementary file 2a* that provides details on intragenic repeats and corresponding sequence for each Flo11p in this study).

**Table 1.** $\beta$ -Aggregation-prone sequences in *Flo11* of various *Saccharomyces cerevisiae* strains*.

|  | Amyloid sequence | Amino acid position | % β-Aggregation |
|---|---|---|---|
| Flo11pL69 | VVSTTVV | 1178; 1285; 1392 | 75.7 |
|  | VTTAVT | 1494 | 59.4 |
| Flo11pBY | VVSTTVV | 1033 | 75.8 |
|  | VTTAVT | 1494 | 84.8 |
| Flo11pΣ | VVSTTVV | 881 | 75.6 |
|  | VTTAVT | 983 | 59 |
| Flo11p133d | VVSTTVV | 1301 | 70.9 |
|  | VTTAVT | 1403 | 59.1 |

*TANGO software (http://tango.crg.es/) with default settings for pH, ionic strength, and temperature was used to determine Flo11p regions with β-aggregation potential superior to 30 %.

Another major difference is the presence of an additional sequence of ~200 amino acids length near the C-terminus of Flo11p$^{L69}$ and which is highy enriched in β-aggregation-prone sequence (*Figure 3A* and see *Supplementary file 3* for TANGO predictor analysis). This additional sequence most likely originated from a twofold repetition of a sequence of about 90–100 amino acids near the C-terminus that is present in all other Flo11p (see *Figure 3*). Interestingly, this repetition led to the presence of two additional amyloid-forming sequences 'VVSTTV' in the Flo11p of strain L69 (*Table 1*).

## Cell wall localization of Flo11p is not altered either by the removal of amyloid-core sequences or by the A-domain or C-terminal of this protein

To investigate the role of the amyloid-forming sequences in the physiological function of Flo11p, we constructed a variant of Flo11p$^{L69}$ in which a region of about 266 amino acids length termed RR2 that contains the two additional amyloid sequences was removed by genome editing of the *FLO11$^{L69}$* gene using CRISPR-Cas9 technology. We also decided to construct additional variants of this protein that lack the A-domain, the C-terminus, and about half of the TRs of the B-domain (*Figure 3A*) to further investigate the role of these domains in the physiological function of Flo11p. All constructs were verified by PCR and by sequencing that the region where the genome editing has been carried out to ensure that it was done in both copies of the *FLO11* gene. According to this approach, the different Flo11p variants were expressed from the endogenous *FLO11* promoter like the wild-type protein, and therefore strains carrying these mutated *FLO11* could be used for direct comparative phenotypic analyses. However, to validate this approach, we had to verify that these genes were indeed expressed, that the encoded protein was present in similar quantities between the different strains and, above all, that the mutated proteins were still located at the cell wall. The expression of the genes encoding the different variants of Flo11p$^{L69}$ was determined by qRT-PCR. As shown in *Figure 3B*, they were expressed at comparable levels, except for the gene encoding Flo11-ΔNter protein, whose transcripts were approximately twofold lower than those of the other transcripts. To determine whether the wild-type and mutated forms of Flo11p are localized in the cell wall, we conducted an immunofluorescence assay using the Alexa Fluor 488-conjugated monoclonal antibody anti-6x-His-tag which cross-reacts with the 6 x-His-tag sequence present at the C-terminus of Flo11p and its variants that have been inserted into the corresponding genes by CRISPR-Cas9 technology as described in Materials and methods. The results of this experiment clearly showed the presence of several fluorescent spots at the cell periphery that correspond to Flo11p localized in the cell wall, in line with previous works (*Fidalgo et al., 2006*; *Guo et al., 2000*). Similar location was observed with Flo11p in which RR1, RR2 region, or even A-domain (Nter deletion) has been removed. More remarkably, removal of the C-terminus that carries the remnant GPI anchor allowing covalent binding of Flo11p to β-1,6 glucans (*Lu et al., 1994*; *Frieman and Cormack, 2003*; *Douglas et al., 2007*) did not abolish the localization of Flo11p to the cell wall (*Figure 4*). Finally, we wished to quantify Flo11p directly in the cell by proteomic analysis. To do so, we prepared cell wall/cell membrane prepared according to *Sarode et al., 2011*. Although several cell wall (i.e. Fks1p, Fks2p, Crh1p, Gas1p, Exg1p, Ygp1p, etc.) and cell membranes proteins (i.e. Pma1p, Hxt6p, etc.) were detected in strain L69, Flo11p as well as other

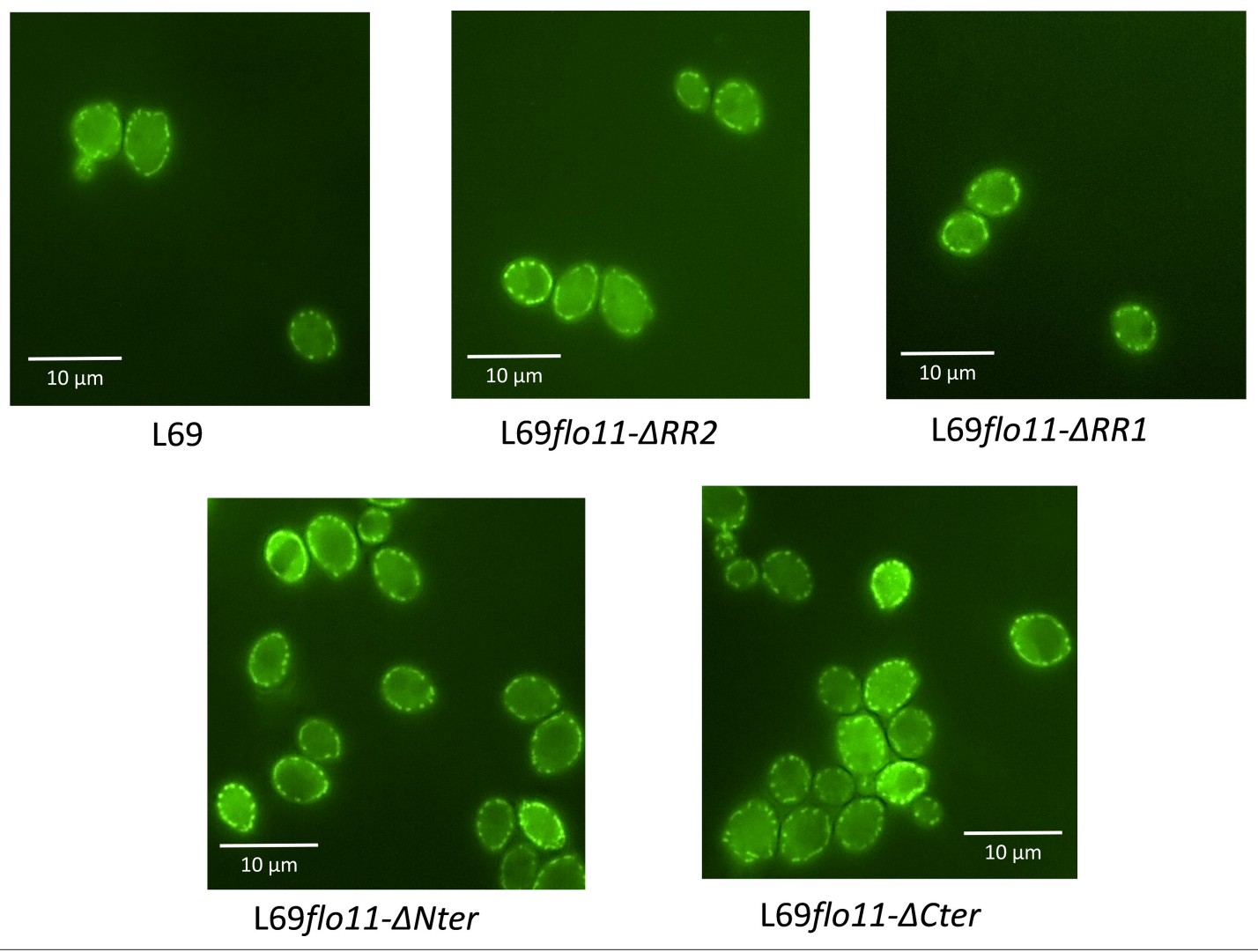

**Figure 4.** The cell wall localization of Flo11p is not impaired by removal of C-terminus, as well as of N-terminus or ablation of RR2 region bearing amyloid-core sequences. Localization of Flo11p and its variants was visualized by confocal microscopy in strain L69 that expressed a 6x-His-tag at the C-terminus of the Flo11 or its variants from the endogenous *FLO11* promoter grown on YPD% and taken at the exponential phase of growth.

flocculins whose expression was validated by transcriptomic were not found (see *Supplementary file 3* reporting a typical proteomic analysis of cell wall/cell membrane sample prepared from L69 strain). In addition, detection of the Flo11p bearing 6x-His-tag at the C-terminus by Western blotting using anti-His antibody in these cell wall/cell membrane preparations was also unsuccessful.

## The amyloid-core sequences in Flo11p from strain L69 are responsible for nanodomains formation

The cell surface of yeast strains expressing the different Flo11 protein variants (i.e. L69*flo11-ΔNter*, L69*flo11-ΔCter*, L69*flo11-ΔRR1*, and L69*flo11-ΔRR2* strain) were imaged by AFM using bare tips. As height images of individual cell trapped in the PDMS microchamber from each of these strains were similar, this argued that the expression of the different Flo11p[L69] variants had no impact on the global surface topology of the yeast cell (*Figure 5A*). From these adhesion images, it can be clearly seen that the ablation of the RR2 region resulted in the inability of the Flo11p variant to produce nanodomains under the AFM tip. In contrast, yeast cells from L69 strain expressing a Flo11p variant lacking either the N- or the C-terminus still showed the formation of nanodomains at the cell surface (*Figure 5*). However, the morphology and nanomechanical properties of these patches were different from those

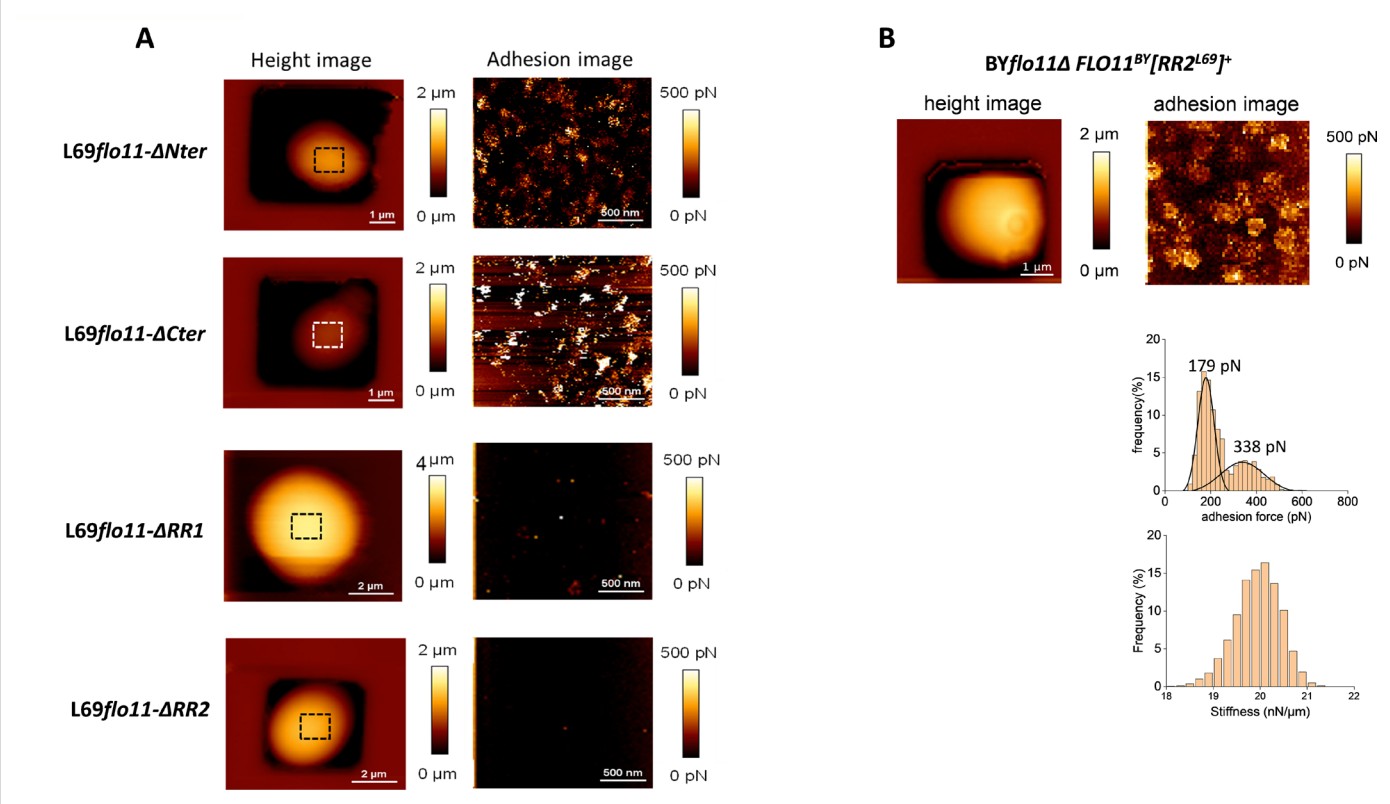

**Figure 5.** Domain in the Flo11p of L69 responsible for the production of adhesive nanodomains. In panel (**A**) are shown atomic force microscopy (AFM) height image (**a**) and adhesion image (**b**) of a single cell from L69 strain expressing Flo11p defective of the N-terminus (*flo11-ΔNter*), C-terminus (*flo11-ΔCter*), or removed from the RR1 (*flo11-ΔRR1*) or RR2 (*flo11-ΔRR2*) domain as depicted in *Figure 3A*. In (**B**) is shown AFM height and adhesion images of a single cell from BY*flo11Δ* strain transformed with pYES2.1 carrying the chimeric *FLO11^BY[RR2]^L69* gene that corresponded to wild-type *FLO11* of BY4741 in which RR2 sequence from *FLO11* of strain L69 has been inserted.

The online version of this article includes the following figure supplement(s) for figure 5:

**Figure supplement 1.** Cell surface analysis of the L69*flo11-ΔNter* strain using silicon nitride (Si3N4) atomic force microscopy (AFM) cantilevers.

**Figure supplement 2.** Cell surface analysis of the L69*flo11-ΔCter* strain using silicon nitride (Si3N4) atomic force microscopy (AFM) cantilevers.

**Figure supplement 3.** Cell surface analysis of L69*flo11-ΔRR1* using silicon nitride (Si3N4) atomic force microscopy (AFM) cantilevers.

obtained by the wild-type Flo11p. Indeed, the nanodomains on the surface of yeast cells expressing the Flo11-ΔNter variant showed a unimodal distribution of low adhesion forces and thus they have lost the high adhesive component (see *Figure 5—figure supplement 1*). Likewise, the patches on the surface of L69*flo11-ΔCter* cells had only the low-adhesion component but they exhibited a needle-shaped morphology with a height that was three times greater than those formed on the surface of L69 cells expressing the normal Flo11p (*Figure 5—figure supplement 2*). Finally, few tiny and disparate patches were noticeable on the surface of L69*flo11-ΔRR1* cell (*Figure 5—figure supplement 3*). These results taken together indicated that the RR2 sequence in the Flo11 protein of L69 strain is essential in the formation of nanodomains but that the other region or domains of Flo11p also contribute to the morphology and to the naonomechanical properties of these nanodomains.

To further ascertain that the RR2 of Flo11p^L69 is determinant in the formation of nanodomains, this sequence was inserted near the C-terminus of the Flo11p of the laboratory BY4741, which otherwise was unable to produce nanodomains even after overexpression of its endogenous Flo11p (see *Figure 2C*). As shown in *Figure 5B*, BY*Δflo11* cells transformed with a multicopy plasmid pYES2.1 carrying this chimeric gene *FLO11^BY[RR2^L69]* under *GAL1* promoter was able to produce prominent highly adhesive nanodomains that exhibited a bimodal distribution of adhesion forces. Altogether, these results confirmed the importance of the RR2 region in the force-induced formation of nanodomains and lends support to the notion that the physical formation of highly and abundant

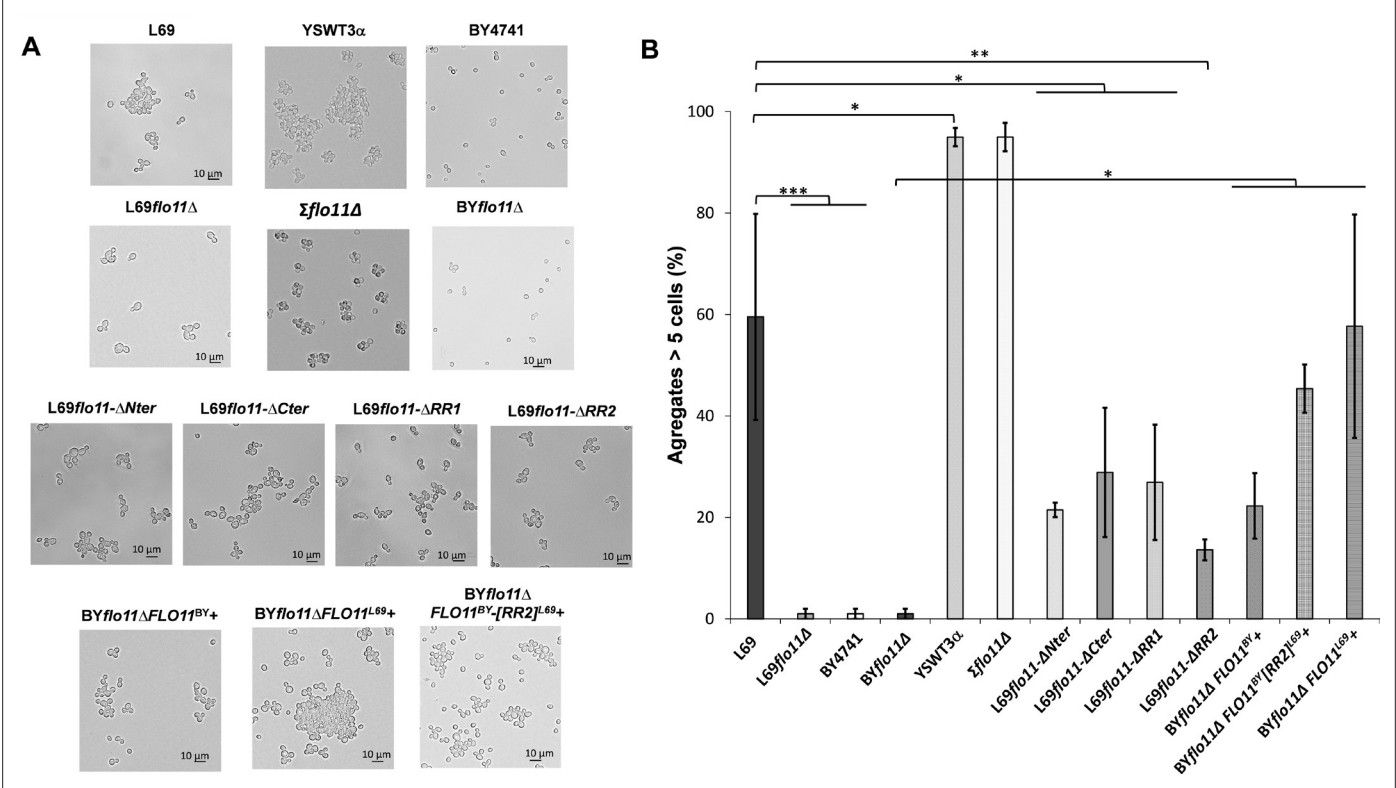

**Figure 6.** The Flo11p-dependent cell-cell aggregation is potentiated by amyloid-forming sequences. The yeast cells were cultivated as in *Figure 5* but until entry in stationary phase and observed under an optical microscope. In (**A**) are shown representative photographs of cell aggregates from the various strains studied. In (**B**) is represented for each strain the percentage of aggregates that are formed by at least five or more cells. For each strain, more than 100 cells or aggregates were counted under the microscope. Values shown are the mean of three biological replicates and vertical bars represent standard deviations. Significant differences are denoted with asterisks (*=p-value < 0.05; **=p-value ≤ 0.01; ***=p-value ≤ 0.005).

The online version of this article includes the following figure supplement(s) for figure 6:

**Figure supplement 1.** Effect of thioflavine S on the Flo11p-dependent cell-cell aggregation.

protuberance at the cell surface of yeast cells required the presence of a threshold number (i.e. at least 3) of amyloid-forming sequence in the Flo11 protein.

## The amyloid-β-aggregation sequences are implicated in the Flo11p-mediated cell-cell interaction

A relevant function of Flo11p is to mediate the formation of different types of multicellular structures such as biofilm, flor, or filament (*Verstrepen and Klis, 2006*; *Brückner and Mösch, 2012*; *Alexandre, 2013*). A common theme to these different phenotypes is cell-cell adhesion or cell-cell aggregation. This phenotype can be easily monitored under an optical microscope by the formation of cell aggregates or clumps (*Purevdorj-Gage et al., 2007*). We therefore quantified this phenotype according to *Purevdorj-Gage et al., 2006* taking into account that an aggregate must contain at least five cells. As shown in *Figure 6A*, both strains L69 and YSWT3a (Σ1278b background strain) nicely exhibited this aggregation phenotype. However, while the deletion of *FLO11* in L69 strain completely abrogated the formation of aggregates, there still remained small clumps of 10–15 cells upon loss of function of this gene in YSTWT3α. As it is reported that *FLO11* is the sole gene of the *FLO* family expressed in this strain (*Guo et al., 2000*), this residual cell-cell aggregation might be due either to another cell wall proteins or to desilencing some silent *FLO* genes through an epigenetic mechanism as described by *Halme et al., 2004*. Whatsoever the mechanism, these data clearly showed that the Flo11p is critical in cell-cell interaction since cells of the laboratory BY4741, which does not express *FLO11* due to non-sense mutation in *FLO8* encoding its major transcriptional activator, do not exhibit aggregation neither.

We then investigated the role of amyloid-forming sequences in this phenotype in carrying out the two following experiments. First, the BY strain deleted of its endogenous *FLO11* was transformed with a high copy pYES2.1 plasmid carrying either its own *FLO11* gene, *FLO11* from L69 strain, or the chimeric gene *FLO11*BY*[RR2*L69*]*, in which RR2 DNA sequence of *FLO11*L69 has been inserted into *FLO11*BY. As reported in *Figure 6A* and quantified in *Figure 6B*, formation of small aggregates of less than 10 cells was observed upon ectopic overexpression of the endogenous *FLO11*BY. However, the size and abundance of these aggregates were dramatically increased in BYΔ*flo11* that overexpresses *FLO11*L69, reaching same capacity of cell-cell aggregation that of strain L69 (*Figure 6B*) and presenting huge aggregates (see *Figure 6—figure supplement 1*). Interestingly, the insertion of RR2 sequence in *FLO11*BY allowed to enhance the formation of cell aggregates as compared to expression of *FLO11*BY (*Figure 6B*). Complementary to this experiment, the role of amyloid-core sequences in cell-cell adhesion could be assessed using amyloid perturbants. We found that aggregates formed in L69 strain or in BYΔ*flo11* that overexpressed *FLO11*L69 were largely disrupted upon incubation with the amyloid disruptor thioflavine S (see in *Figure 6—figure supplement 1*), whereas, as expected, this drug had no effect in strain lacking Flo11p (strain L69 Δ*flo11*, data not shown) or in strain L69*flo11-ΔRR2* that expressed a Flo11p variant lacking amyloid-forming sequences (*Figure 6—figure supplement 1*). On the contrary, cell-cell aggregation in YSWT3α strain was not disturbed upon incubation with thioflavin S (see *Figure 8—figure supplement 1*), indicating that the phenotype which is otherwise dependent to a large extent on Flo11p (see data in *Figure 6A*) does not imply amyloid-forming sequences. Collectively, these results indicated that amyloid-core sequences contribute to cell-cell adhesion but are not essential for this phenotype.

To further illustrate the contribution of amyloid-core sequences in cell-cell adhesion, we removed the RR2 region from Flo11pL69 and found that this ablation resulted in a 75 % reduction of the cell-cell interaction, and this reduction was statistically higher than that of the removal of the A-domain (Flo11-ΔNter) (*Figure 6A and B*). The contribution of the other domains of Flo11p to this phenotype was also evaluated. The percentage of cell aggregates dropped by approximately 50 % upon deletion of the C-terminus, which could be accounted by the loss of two out of the four amyloid-forming sequences

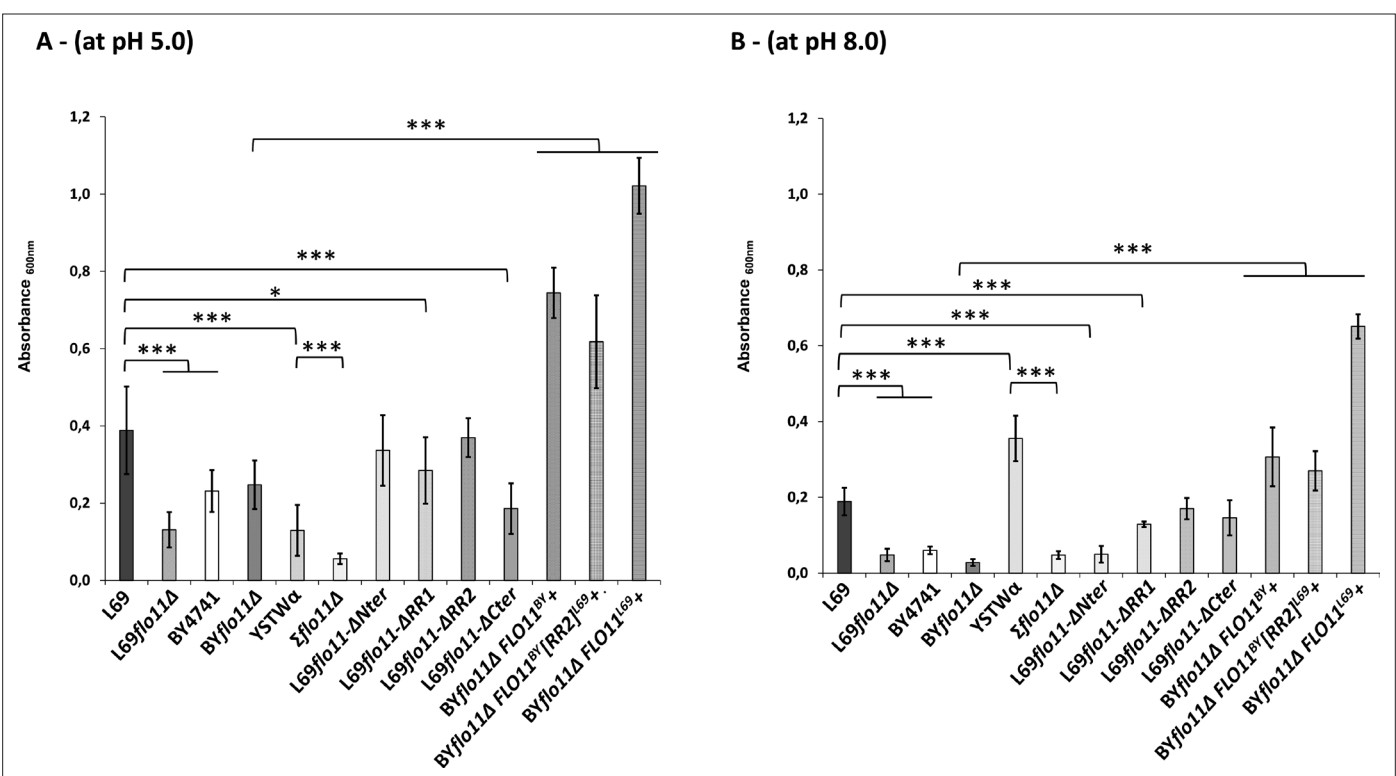

**Figure 7.** Adherence to plastic is strongly impacted by pH and does not solely depend on the A-domain of Flo11p. Adherence of yeast cells was carried out a 96-well polystyrene plate as described in Materials and methods. The data are the mean of three replicates measurements ± standard deviation. Significant differences are denoted with asterisks (*=p-value < 0.05; **=p-value ≤ 0.01;***=p-value ≤ 0.005).

(see *Figure 3*). A 50 % reduction of cell aggregates was also observed in the strain that expresses Flo11p lacking part of the B-domain (RR1 sequence). Altogether, these results indicated that the in vivo efficiency of cell-cell interaction requires the full Flo11 protein.

## Investigation of domains of Flo11p required for adherence and surface hydrophobicity properties

The contribution of the amyloid-core sequences and of various Flo11p domains on adherence and hydrophobicity properties was then assessed using the dedicated strains constructed above, which expressed Flo11p variants from the endogenous *FLO11* promoter. Adherence of cells as assayed on polystyrene surface (*Reynolds and Fink, 2001*) was, as reported earlier (*Kraushaar et al., 2015*), strongly affected by pH (*Figure 7*). However, this pH effect was not comparable between yeast strains. While adherence to plastic of strains L69 and BY4741 was significantly reduced by raising the pH from 5.0 to 8.0, the opposite was found for strain YSTWα. However, this property did not rely exclusively on the presence of Flo11p since this phenotype was observed in BY4741 strain that does not express this protein (*Liu et al., 1996*), whereas the deletion of *FLO11* in strain L69 and YSTWα did not completely abrogate adherence to polystyrene. On the other hand, our data showed that the potency of cells to adhere to plastic was dramatically enhanced by overexpression of *FLO11* as found with BYΔflo11 cells that ectopically overexpressed this gene from a high copy plasmid carrying *FLO11* under *GAL1* promoter (*Figure 7*). However, it is worth noticing that cells overexpressing *FLO11*[L69] had higher adherence to plastic than those expressing *FLO11*[BY], whether the assay was carried out at pH 5.0 or 8.0 (p-value < 0.005). This more potent effect of *FLO11*[L69] could probably due to the strongest

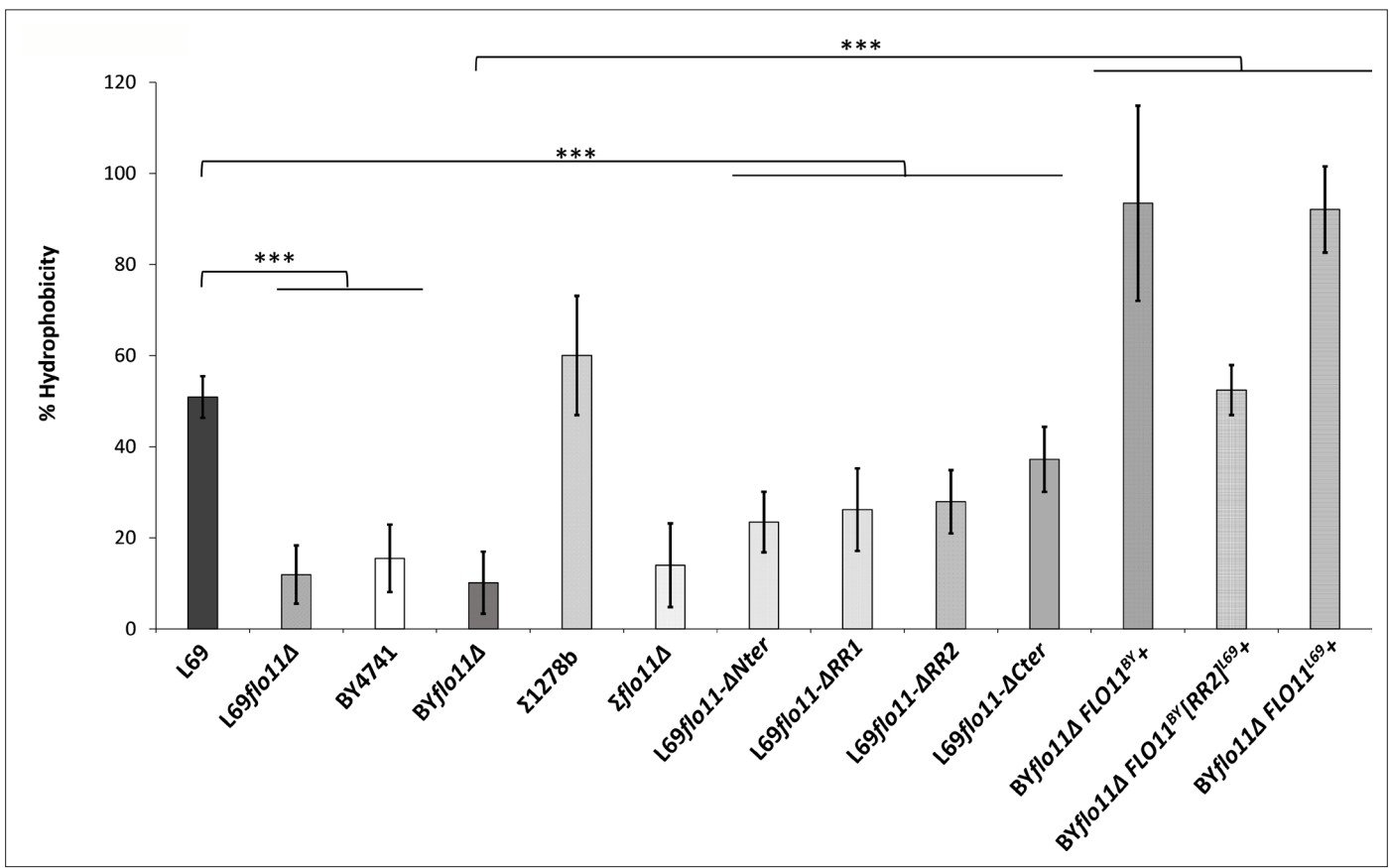

**Figure 8.** Surface hydrophobicity mainly depends on abundance of Flo11p. Surface hydrophobicity corresponded to the percentage of cells partitioning in the octane layer, as described in Materials and methods. The data are the mean of three biological replicates and vertical bars represent standard deviations. Significant differences are denoted with asterisks (*=p-value < 0.05; **=p-value ≤ 0.01;***=p-value ≤ 0.005).

The online version of this article includes the following figure supplement(s) for figure 8:

**Figure supplement 1.** Assay of velum formation.

expression of this gene compared to $FLO11^{BY}$ in the BYΔflo11 cells rather than to the additional presence of amyloid sequences in Flo11p[69] as indicated by our transcript analysis (see *Figure 2—figure supplement 1*). In addition, the overexpression of the chimeric construct $FLO11^{BY}$-$RR2^{L69}$ in BYflo11Δ cells resulted in a similar adherence as overexpression of $FLO11^{BY}$. Further analysis of the domains of Flo11p that are implied in adherence to polystyrene showed that the C-terminal domain contributed the most to this property at pH 5.0, whereas at pH 8.0, the removal of the N-terminus of Flo11p$^{L69}$ resulted in a drop of adherence equal to that obtained upon deletion of *FLO11* gene. This result is, on the one hand, in line with the model of *Kraushaar et al., 2015* that the N-terminus of Flo11p is essential for cell adhesion to polystyrene, but on the other hand, it did not support their claims that this mediation is only effective at low pH. To sum up, these results indicate that the rate of cell adherence to plastic is dependent on the amount of Flo11p produced on the cell surface while the ability to adhere to these surfaces is influenced by the sequence of this protein.

Surface hydrophobicity was evaluated as the percentage of yeast cell partitioned in the octane phase as described in previous work (*Purevdorj-Gage et al., 2006*; *Van Mulders et al., 2009*). Results in *Figure 8* show that it is mainly the amount of Flo11p present at the cell surface that determines this property as indicated by the significant increase in hydrophobicity of cells that overexpressed *FLO11* regardless the origin of the gene. This property was not entirely dependent of Flo11p as indicated by the residual hydrophobicity in strains deleted for *FLO11* (L69Δflo11 and ΣΔflo11) or that cannot express this gene (i.e. BY4741). Our data also showed that each domain of Flo11p contributed to this phenotype, with the N-terminal carrying the higher contribution and the C-terminal the lowest. Since surface hydrophobicity is critical in the formation of buoyant biofilm that is strictly dependent on Flo11p (*Fidalgo et al., 2006*), we examined whether L69 strain, which is a wine yeast, could exhibit this phenotype (*Alexandre, 2013*). Contrary to expectation, neither L69 strain nor strain expressing any of the Flo11p variant had the ability to form this air-liquid biofilm (termed also velum) at the surface of the culture, whereas this phenotype was nicely visualized in the flor strain A9 as described in *Zara et al., 2005* and *Figure 8—figure supplement 1*. Taking into account that the number of repeats in the B-domain is of paramount importance in velum formation according to *Fidalgo et al., 2006*, it is at first sight difficult to credit the absence of this phenotype in strain L69 to a lack of these repeats, since there are as many repeats in this protein as there are in the Flo11p of strain 133d (46 vs. 49, see *Supplementary file 2*).

## The Flo11p-dependent invasive growth in agar requires N- and C-terminal of the protein, but its elicitation and intensity depend on other factors that are defective in S288c background strain

It is well established that *FLO11* gene is required for pseudohyphal development in diploids and for invasive growth in haploids strains of *S. cerevisiae* (*Gancedo, 2001Gancedo, 2001*), although this invasive phenotype can be provoked in diploid strains upon overexpression of this gene (*Lo and Dranginis, 1998*). This phenotype is commonly assessed by agar invasion assay that consists in cultivating yeast cells on agar plates containing rich or synthetic sugar medium for several days and then examining those cells that remained sticky on the agar plates after extensive washing under a stream of water (*Roberts and Fink, 1994*). Applying this assay to L69 strain, we found that this strain exhibited massive invasion in agar on both rich (YPD) and synthetic (YNGal) sugar medium (*Figure 9*). The finding that invasiveness on agar occurred in this strain, which is diploid, may be attributed to the relatively high expression of *FLO11* measured in this strain (*Schiavone et al., 2015*). As expected, agar invasion was also observed for the haploid YSWT3αstrain, a derivative of Σ1278b (*Dowell et al., 2010*), whereas it was totally absent in BY4741 strain, likely because *FLO11* is not expressed in this strain (*Liu et al., 1996*). Accordingly, agar invasion was lost upon deletion of *FLO11* in L69 and YSTW3α strains, which confirmed the critical function of this gene for this phenotype (*Lo and Dranginis, 1998*). As already noticed by *Guo et al., 2000*, invasion in agar was less pronounced on agar plates made with a galactose medium (see *Figure 9—figure supplement 1*). In addition, we found that cells that remained more sticky after washing are those at the periphery of the spot when the invasion experiment was carried out in a rich sugar medium, whereas cells at the heart of the spot showed the most invasiveness in a synthetic medium (*Figure 9* and see also *Figure 9—figure supplement 1*). More surprisingly, the diploid Σ1278b and isogenic derivative haploid YSTW3α were unable

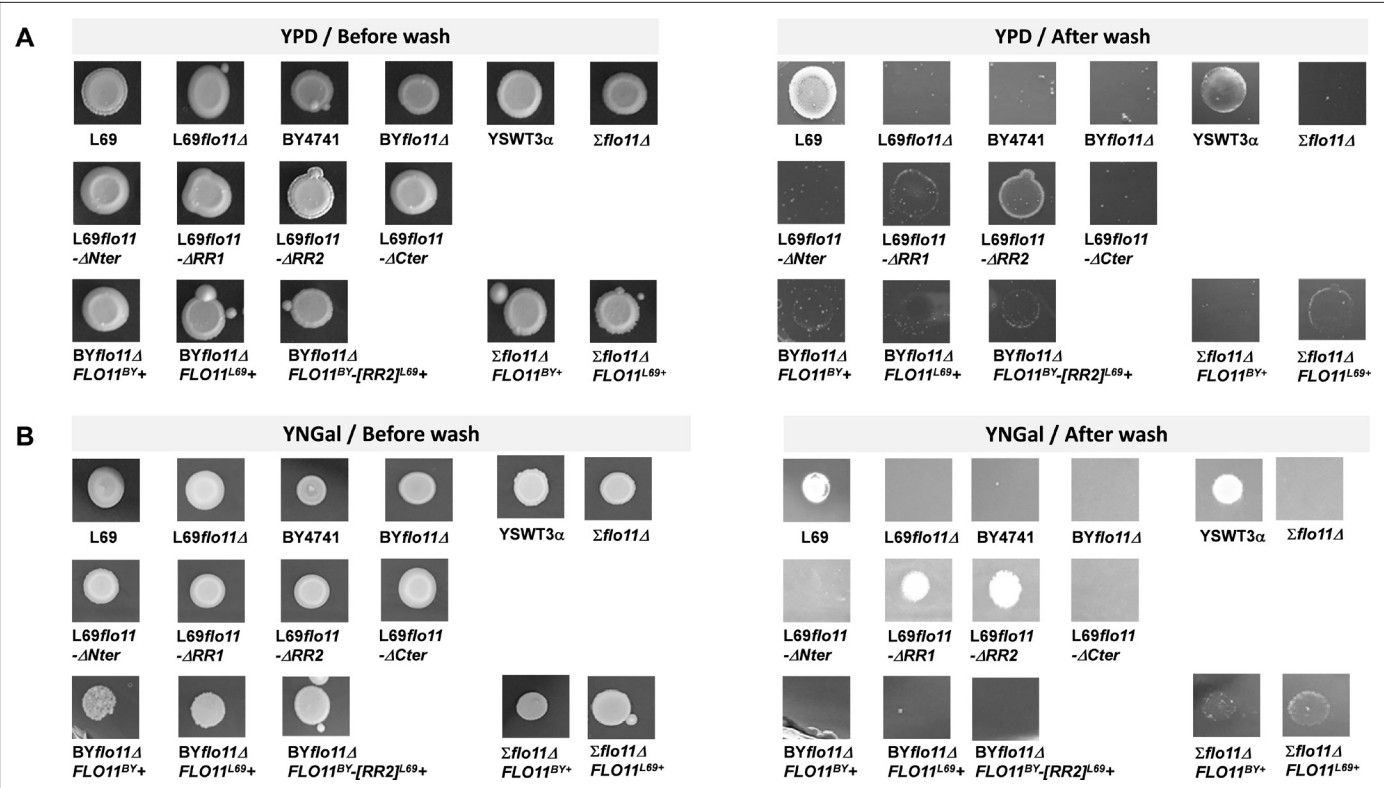

**Figure 9.** The Invasive growth in agar is abolished by deletion of either N- or C-terminal of the Flo11p. All strains were pre-grown in YNGal that was supplemented with the auxotrophic requirements when needed (i.e. uracil, leucine, histidine, methionine at 0.1 % for BY4741, BY*flo11Δ*, and YSTW3α but only leucine, histidine, methionine for BY*flo11Δ* and Σ *flo11Δ* expressing *FLO11^BY^*, *FLO11^L69^*, or *FLO11^BY^-[RR2]^L69^*) until stationary phase. Then, 10 µl of these cultures were spotted on 2 % (w/v) agar plate made with YPD (panel A) or YNGal complemented with auxotrophic requirements (panel B). Plates were incubated at 30 °C for 8 days and washed under a stream of water. They were photographed before and after washing.

The online version of this article includes the following figure supplement(s) for figure 9:

**Figure supplement 1.** Invasive growth in agar by different yeast strains, and impact of the culture medium and of the domains of FLO11p on this phenotype.

**Figure supplement 2.** Sensitivity of L69 and L69 mutant strains to caffeine and calcofluor white (CFW) drugs.

to elicit an invasion growth phenotype in a synthetic glucose medium, whereas this capacity was still very effective in L69 strain (see *Figure 9—figure supplement 1*).

We then investigated which domains or sequences of the Flo11p^L69^ are necessary for the in vivo invasive growth in agar. Results in *Figure 9* show that this phenotype was completely lost in strains expressing a Flo11p that lacks either the N- or the C-terminal domain, reduced in L69*flo11-ΔRR1* cells and weakly affected upon the removal of the RR2 sequence (*Figure 9* and see also *Figure 9—figure supplement 2*). More puzzling results were obtained with the laboratory strain BY4741 transformed with *FLO11^L69^* or *FLO11^BY^* carried on a 2µ plasmid under the *GAL1* promoter (pYES2.1). Indeed, we found that this strain failed to invade agar in spite of huge ectopic overexpression of *FLO11^BY^* or *FLO11^L69^* as in YNGal plates lacking uracil (YNGal Ura⁻) (*Figure 9*) under which these genes were shown to be exceedingly transcribed (*Figure 3—figure supplement 1*). On the other hand, the invasiveness of the haploid YSWT3α deleted for *FLO11*(Σ flo11Δ) transformed with pYES2.1 plasmid carrying either *FLO11^BY^* or *FLO11^L69^* was clearly discernable in the selective YNGal Ura⁻ (*Figure 9*), with *FLO11^L69^* being slightly more effective than *FLO11^BY^*. Nonetheless, the invasive phenotype of Σ flo11D that overexpressed *FLO11^BY^* or *FLO11^L69^* was weaker than that of this strain expressing its endogenous Flo11 protein.

## Flo11 protein and the cell wall integrity

Flo11p is a highly mannosylated cell surface protein that is retained to the cell wall inner network of β-1,6 glucans through a GPI anchor (*Klis et al., 2006*). We found that the loss of function of this gene

resulted in a higher sensitivity to drugs such as calcofluor white (CFW), caffeine, and Congo red (CR) that are commonly used to assess the integrity of the yeast cell wall (*Levin, 2005*). Furthermore, we found that the higher sensitivity to these cell wall drugs was mainly associated with the lack of the C-terminal of Flo11p although removal of RR1 or RR2 sequence of Flo11$^{L69}$ protein resulted in mutant strains that were also slightly more sensitive than wild type (see *Figure 9—figure supplement 2*). Under, glycosylation of Flo11p could be invoked for the slight increase of sensitivity to cell wall drugs of strains expressing variants of Flo11$^{69}$p that have been deleted for RR1 since the repeated sequences in this region are thought to be heavily glycosylated and cell surface glycosylation is important in the cell wall integrity (*Strahl-Bolsinger et al., 1999*; *Free, 2013*). On the other hand, the higher sensitivity of L69*flo11-ΔCter* strain could not be due to a lack of retention of the protein at the cell surface due to the removal of the GPI anchor, since our immunofluorescence assay indicated that this Flo11p variant lacking this domain still localized at the cell surface.

## Discussion

In this work, we showed that the abundant patches observed by AFM on the cell surface of the industrial wine yeast L69 strain (*Schiavone et al., 2015*) correspond to adhesion nanodomains, with nanomechanical properties very similar to those formed on the cell surface of the pathogen *C. albicans* (*Alsteens et al., 2010*; *Formosa et al., 2015b*). Moreover, we reported that the flocculin encoded by *FLO11* is the sole protein involved in the formation of these nanodomains, which can be prevented upon treatment of the cells with anti-amyloid peptides or anti-amyloidophilic dyes. Altogether, these data are in line with the model of *Lipke et al., 2018*; *Lipke et al., 2012* arguing that adhesion nanodomains require cell wall proteins that must harbour in their sequence the following features: (i) short amyloid-β-sheet interaction sequences of five to seven amino acid residues and (ii) serine/threonine-rich 'T domain' enriched of β-branched aliphatic amino acids Ile, Val, and Thr. While all flocculins encoded by *FLO1*, *FLO5*, *FLO9*, and *FLO11* in *S. cerevisiae* display these features, it was intriguing to know why such dense and abundant nanodomains had not been observed so far in this yeast species so far, even though physical *FLO*-dependent modification of cell surface upon hydrodynamic shear force has been recorded (*Chan and Lipke, 2014*; *Chan et al., 2016*). The data reported in this work clearly showed that the Flo11 protein of the *S. cerevisiae* strain L69 differs from that of other *Saccharomyces* strains sequenced to date by having an additional sequence of ~200 amino acids length (RR2 sequence) near the C-terminus. This sequence probably originates from a twofold repeat of a 90–100 amino acids sequence present near the C-terminus of all Flo11 flocculins that contains the amyloid-forming sequence 'VVSTTV' followed by 'ITTTFV'. This repeat provides therefore two additional amyloid-core sequences to the Flo11p of strain L69, which are clearly determinant in the process of nanodomains formation. Indeed, we showed that the insertion of this RR2 sequence into the Flo11p of BY4741 allowed this strain to form adhesion nanodomains with nanomechanical properties similar to those formed on the cell surface of strain L69.

Our AFM analysis revealed that these nanodomains encompass two components: a first one characterized by a high adhesion force, which corresponds to amyloid nanodomains as they are disrupted under the action of anti-amyloid agents, while the other component has low adhesion force which likely corresponds to isolated Flo11 proteins and possibly other cell wall proteins that may unfold under the action of the AFM tip. This behaviour is fully reminiscent to the nanodomains formed by adhesins on the cell surface of the pathogen *C. albicans* (*Formosa et al., 2015bFormosa et al., 2015b*). This analysis also highlighted that, while the amyloid-core sequences are essential in the formation of adhesion nanodomains, other parts of the Flo11p sequence also contribute to their production. In particular, the removal of about half of the TRs in the B-domain almost completely abolished nanodomains formation, while the highly adhesive component of these nanodomains was no longer observed after removal of the N- or C-terminus of Flo11p. These results indicated that the whole Flo11 protein is needed to acquire the right production of adhesion nanodomains and support the model of *Lipke et al., 2012* suggesting that upon the traction of an AFM tip or any other physical stretch, the adhesins are unfolded through their TRs, which expose amyloid-core sequences for interaction between neighbouring molecules.

The existence of the unique RR2 sequence in Flo11p of strain L69 led us to investigate the role of amyloid-forming sequences in the physiological function of this protein in relation to the role of other domains of this protein. For this purpose, we used CRISPR-Cas9 technology to create Flo11p variants

specifically defective in these domains or sequences and whose expression is strictly dependent on the endogenous *FLO11* promoter. This approach should allow us to perform comparative phenotypic analyses under conditions where the expression of these proteins is physiologically comparable. While our qRT-PCR fully confirmed this assertion, proteomic analysis carried out on cell wall/cell membrane preparation from L69 strain failed to detect Flo11p although other relevant cell wall proteins were identified. A likely explanation of this failure is the fact that Flo11p as well as other flocculins are not abundant cell wall proteins. Nevertheless, immunofluorescence experiments enabled to clearly identify Flo11p at the cell periphery and remarkably, same localization was found for all different Flo11p variants, leading to conclude that they are all in the cell wall. This result calls into question the fact that the modified GPI anchor at the C-terminus of Flo protein is essential for these flocculins to be retained at the cell wall (*Bony et al., 1997*; *Kapteyn et al., 1999*; *Douglas et al., 2007*). However, the finding of Flo11 protein in the extracellular medium when the GPI anchor was removed (*Douglas et al., 2007*) could be explained by the fact the corresponding gene was expressed from a high copy plasmid under the strong *PGK1* promoter, which likely resulted in the production of a huge amount of protein that exceeds its ability to be entirely retained in the cell wall. Our immunofluorescence assays are also consistent with AFM data suggesting that Flo11 variants lacking C- or N-terminus are expressed at the cell surface, although firm demonstration of this cell surface localization should require further experiments using ligands, such as antibodies, recognizing specifically these truncated forms of Flo11p.

Cell-cell adhesion is a major cell wall property of Flo11p which is translated into either the sedimentation of cells as flocs at the bottom of the culture medium (flocculation phenotype), the formation of biofilm at the air-liquid interface (flor phenotype), or even biofilms on abiotic surface (adherence to plastic) (*Verstrepen and Klis, 2006*; *Brückner and Mösch, 2012*; *Alexandre, 2013*). Based on their work, Mösch and co-workers (*Kraushaar et al., 2015*; *Brückner et al., 2020*) proposed a Flo11p-dependent model of cell-cell adhesion that can furthermore allow for discrimination between cells from the same and from a different kind (kin discrimination). In a first model, it is proposed that Flo11 molecules cluster in *cis*- via amyloid β-sheet interactions sequences followed by interactions in *trans* between clusters of Flo11p of opposing cells. An alternative model was to suggest a simple bi-molecular Flo11p-Flo11p *trans*-interaction. Whatever the model, these authors showed an essential role of the A-domain, which encompasses 185–240 aa length at the N-terminal after the secretion signal of Flo11p, in cell-cell adhesion and indicated that this interaction is promoted at acidic pH and reduced at alkali pH. While our data did not reproduce very well those from *Kraushaar et al., 2015* with respect to the importance of A-domain in the formation of cell aggregates and adherence to polystyrene, they however could support the models of cell-cell adhesion proposed by these authors. Indeed, the removal of amyloid-forming sequence in Flo11p of the L69 strain resulted in 70 % drop in cellular aggregation, whereas loss of Flo11p abrogated this phenotype, indicating that the remaining aggregation can be due to simple bi-molecular interactions of isolated Flo11p between cells. Conversely, cell-cell interaction was reduced by 50 % in a strain expressing Flo11p defective of the A-domain, suggesting that amyloid-forming sequences could in part overcome this defect and thus mediate interaction in *trans*. The finding that cell-cell aggregation was higher in BY4741 expressing a chimeric Flo11[BY]-RR2[L69] protein than in the same strain expressing its own endogenous Flo11p also supports this suggestion, as well as the fact that huge aggregates seen in BY4741 strain overexpressing *FLO11[L69]* are cancelled by incubation with anti-amyloidophilic agents. Collectively, our results could be put in perspective with the recent work of Lipke and colleagues (*Ho et al., 2019*; *Dehullu et al., 2019a*) which used an innovative fluidic force microscopy enabled to manipulate and measured force at the single cells level to show that homotypic interactions between opposing cells can be mediated by amyloid-forming sequences. Nevertheless, the implication of amyloid-forming sequence is likely not an absolute requirement for cell-cell aggregation as witnessed by the finding that strain Σ1278b, which is known to only express Flo11p, forms huge aggregates that are resilient to the treatment with anti-amyloidophilic agents. These data could therefore agree with a model of Flo11p *trans*-interaction only mediated by A-domain.

Another dominant phenotype brought about by the expression of *FLO11* is the ability of haploid yeast cells to undergo invasive growth in agar or pseudohyphal growth for a diploid cell (*Gancedo, 2001*). In this work, we found that strain L69 exhibited a very intense invasive growth phenotype, which was at first glance surprising since this strain is diploid. This phenotype is commonly not expressed in diploid cells unless *FLO11* is highly expressed (*Lo and Dranginis, 1998*). However, it is unlikely that

this pronounced agar invasion of L69 strain was solely due to high expression of *FLO11* since our proteomic analysis failed to detect Flo11p. In addition, it should be noticed that the so-called high expression of *FLO11* in L69 strain was made in reference with BY4742 strain that does not express this gene (*Schiavone et al., 2015*). Therefore, other yet unclear mechanisms must exist in this strain to account for this peculiar phenotype. Beside this open question, we showed that the N- and C-terminus of Flo11p are indispensable to elicit agar invasion phenotype, whereas amyloid-forming sequences have no role in this process. Our data are consistent with those of *Kraushaar et al., 2015* who previously showed that the deletion of the A (N-terminal) domain of Flo11p abrogates agar adhesion, and also highlight a role for the C-terminus in this phenotype. This latter result could be explained by the inability of the cell to remain trapped in agar due to the fact that Flo11p, in the absence of its C-terminus, is still localized at the cell wall as indicated by our immunofluorescence assay, but loosely retained in this external structure. The invasive growth in agar was also strongly diminished upon deletion of repeat sequences in the B-domain, which can be explained by a reduction of glycosylation of Flo11p as a defect in this process was found to reduce invasive growth in agar (*Meem and Cullen, 2012*). Alternatively, the removal of TR in B-domain (i.e. RR1 sequence) may prevent Flo11p from reaching the cell surface and therefore reduce the capacity of the cells to remain trapped in the agar. Finally, this study brought to light new data on the process of invasive growth. On the one hand, we found that invasion in agar of strain L69 was equally intense under any kind of growth media, whereas this phenotype in Σ1278b background strain was weak in a galactose agar medium and absent in a glucose agar synthetic medium. In addition, the potent agar invasion phenotype of strain L69 could not be solely attributed to *FLO11* gene because the overexpression of this gene in YSTW3a *flo11Δ* (derivative of Σ1278b) did not provide to this strain a higher capacity to invade agar than that of the isogenic YSTW3α expressing its endogenous gene. More puzzling was to find that the ectopic overexpression of *FLO11* in BY4741 did not rescue the inability of this haploid strain to elicit invasive growth phenotype. Altogether, these data indicated that the Flo11p-dependent invasive phenotype involves additional genetic factors that are absent in BY4741 and that are more expressed in L69 strain, accounting for the higher potency of this strain to invade agar.

Based on this work and on several previous works on Flo11p in the yeast *S. cerevisiae*, one can raise the question why strains that naturally express Flo11p or in which this gene has been ectopically expressed do not exhibit collectively all these phenotypes of flocculation, flor, or growth invasion. Clearly, a large part of the answer lies in the sequence of this protein, and more precisely in differences found in the three domains of this protein, as already suggested by *Barua et al., 2016* for the phenotypic comparison between Flo11p from *S. cerevisiae* var *diastaticus* and Σ1278b. As an example, the pH dependence and adherence to polystyrene is totally different between L69 and Σ1278b strain. This difference could be explained by the fact that Flo11p in the latter strain has a 15 amino acid insertion in the N-terminal at position 115–130 that was shown to strongly enhance the adhesive property of Flo11p (*Brückner and Mösch, 2012*). Likewise, the ability of Flo11p to elicit a flor phenotype in yeast has been attributed to high copy number of TRs in the B-domain, which enhances surface hydrophobicity via higher glycosylation (*Fidalgo et al., 2006*). However, the strain L69 does not exhibit flor phenotype although the expansion of the B-domain in Flo11p of this strain is very similar to that of strain 133d (46 TRs vs. 49 TRs). The lack of flor phenotype of L69 strain could be due either to a lower glycosylation of its Flo11p or to the type of TRs sequences, which in the case of Flo11p of this strain is represented by four different sequences, while only one single sequence of 81 nt is repeated 49 times in the Flo11p of strain 133d. In summary, the Flo11p sequence has been subjected to evolution to adapt yeast to its specific ecological niche and, therefore, the acquisition of additional amyloid-forming sequences in the Flo11p of the L69 strain could be due either to its original niche or to the particular conditions related to the industrial process for its propagation.

## Materials and methods
### Strains and growth conditions

The *S. cerevisiae* strains used and constructed in this work are listed in *Supplementary file 4*. Unless otherwise stated, strains were cultivated in rich YPD medium at 30 °C (1% w/v yeast extract, 1% w/v bactopeptone, and 2% w/v glucose), When using synthetic YNGlu or YNGal media (0.17% w/v yeast nitrogen base, 0.5% w/v $(NH_4)_2SO_4$, with 2% w/v glucose or 2% w/v galactose), they were supplemented

with appropriate amino acids at 0.1 % (w:v). YNB acetamide medium (0.17% w/v yeast nitrogen base, 0.66% w/v K$_2$SO$_4$, 0.06% w/v acetamide, and 2% w/v glucose) was used for the selection and propagation of L69 mutants transformed with plasmid bearing amdSYM cassette (*Solis-Escalante et al., 2013*) used for CRISPR-Cas9 deletion (*Ryan et al., 2016*). YNB fluoroacetamide medium (0.17% w/v of yeast nitrogen base, 0.5% w/v of (NH$_4$)2SO$_4$, 0.23% w/v fluoroacetamide, and 2% w/v glucose) was used to excise the amdSYM cassette from the genome of L69 generated mutants. For solid media, agar was added at 2 % (w/v) before sterilization at 120 °C for 20 min.

## Plasmids and strains construction

To overexpress *FLO11* gene or any of its alleles in yeast strain, expression vector pYES2.1 TOPO TA (*Supplementary file 5*) was used which carries *GAL1* as promoter and *CYC1* terminator. The *FLO11* ORF from BY4741 (*FLO11*$^{BY}$) and L69 (*FLO11*$^{L69}$) were amplified by PCR using the primers FLO11_TOPO_f and FLO11_TOPO_r (*Supplementary file 6*) and ligated into pYES2.1 TOPO TA vector (Thermo Fisher Scientific) between *GAL1* promoter and *CYC1* terminator according to the manufacturer's protocol. Correct integration was confirmed by restriction digestion and plasmid sequencing.

Deletion of *FLO11* was constructed in L69 strain using CRISPR-Cas9 strategy (*Ryan et al., 2016*). The high copy pCas9-amdSYM plasmid (derived from pML107, Addgene) that constitutively expresses the gene encoding Cas9 endonuclease protein and carrying a gRNA expression cassette (Sap1 cloning sites) was used. It confers the ability to *S. cerevisiae* to use acetamide as the sole nitrogen source (*Solis-Escalante et al., 2013*) enabling selection of prototroph strains. As the CRISPR/Cas9 technique requires the identification of a unique 20 N (NGG) sequence, the candidate target sequence was identified using the online software CRISPR-direct (https://crispr.dbcls. jp) and determined as to be located in the middle of each region of *FLO11* to be deleted. Linear healing fragments of 120 bp were designed with 60 bp overlapping the upstream and downstream sequences of the region to be deleted, namely the N-terminal (from 4 to 675 bp), the C-terminal (from 3889 to 5163 bp), the RR1 domain (from 735 to 2573 bp), and the RR2 domain (from 3575 to 4375 bp). Ligation of gRNA sequence into the plasmid was made using a T4 DNA ligase at 16 °C overnight and transformed into DH5α chemically competent bacteria according to the manufacturer's protocol (NEB). The presence of the new gRNA into pCas9-amdSYM was confirmed by Sanger sequencing using the M13 forward primer. The generated plasmids were used to transform L69 strain and healing fragments were added to repair the double strand break made by CRISPR-Cas9 protein according to the wanted deletion. Constructions were verified by PCR amplification and Sanger sequencing (*Supplementary file 6*).

For BY*flo11Δ FLO11*$^{BY}$*-[RR2]*+ strain, a chimeric gene constituted of the N-terminal and central regions of *FLO11* from BY4741 strain fused to the C-terminal domain of *FLO11* from L69 (including the RR2 region) was constructed. To this end, a 2967 bp fragment was PCR-amplified from *FLO11*$^{BY}$ using the primers FLO11_TOPO_f and FLO11_BY_2964_r (*Supplementary file 6*), whereas a 1794 bp fragment was PCR-amplified from *FLO11*$^{L69}$ using the primers R269 inFLO11BY_f and FLO11_TOPO_r (*Supplementary file 3*). Both fragments share a 30 nucleotides sequence overlapping that allow their fusion by overlapping PCR. The resulting 4731 nucleotides fragment was PCR-amplified using FLO11_TOPO_f and FLO11_TOPO_r oligonucleotides (*Supplementary file 6*). The resulting chimeric gene *FLO11*$^{BY}$*-[RR2]*$^{L69}$ was ligated into pYES2.1 TOPO TA vector as described above.

To examine localization of the various Flo11 variant proteins by immunofluorescence, a 6x-His-tag was added at the C-terminal (before stop codon) of these variants as well as the wild-type Flo11p using CRISPR-Cas9 strategy as described before, with plasmid pCas9-amdSYM carrying same RNA guide (SAPI_ARNg_HIStag_L69) used in all cases. Healing fragments used in *FLO11Δ*-Cter were purchased from Eurofins (*Supplementary file 6*), whereas healing fragments used in *FLO11* WT, *FLO11-ΔNter*, *FLO11-ΔRR1*, and *FLO11-ΔRR2* were constructed by PCR amplification of *FLO11* using four primers (*Supplementary file 6*). A first 534 pb amplicon was obtained using PCR-HF_L69_fw1 and PCR-HF_L69_rv1 primers and a second one of 537 pb using PCR-HF_L69_fw2 and PCR-HF_L69_rv2 primers (*Supplementary file 6*). Healing fragment of 1071 pb was finally obtained by PCR amplification with PCR-HF_L69_fw1 and PCR-HF_L69_rv2 primers using a mix of both resulting amplicons as DNA matrix. All gene constructs were verified by sequencing.

Yeast strains were then transformed with each of the resulting plasmids using yeast transformation procedure as described in *Gietz and Schiestl, 2007*.

## Bioinformatics

The gene *FLO11* was retrieved from the L69 strain genome which has been fully sequenced internally (Lallemand, Inc, unpublished data) using the Pacific Biosciences method, whereas the sequences of *FLO11* in the laboratory strain BY4741 and $\Sigma$1278b were uploaded from the SGD database (https://www.yeastgenome.org). Clustal Omega was used to perform amino acid sequence alignments (*Sievers and Higgins, 2014*). Structural and functional comparison between the sequences of Flo11p$^{L69}$, Flo11p$^{BY4741}$, and Flo11p$^{\Sigma1278b}$ were carried out using HCA (*Lemesle-Varloot et al., 1990*) giving a plot of each open reading frame that draws as a helical projection, vertically repeated. Secretion signal and the GPI signal anchorage to cell wall β-glucan were searched using SignalP-4.1 server (*Nielsen, 2017a*, *Nielsen, 2017b*), PredGPI tool (*Pierleoni et al., 2008a*, *Pierleoni et al., 2008b*). TANGO software (*Fernandez-Escamilla et al., 2004a*, *Fernandez-Escamilla et al., 2004b*) with default settings for pH, ionic strength, and temperature was used to determine Flo11p regions with β-aggregation potential superior to 30 %. Intragenic repeats in the *FLO11* ORF were screened using the EMBOSS ETANDEM software (*Rice et al., 2000*, *Durbin et al., 2000*). Criteria were selected by those with a repetition length >10, a score >20, and a repetition conservation >60%.

## Atomic force microscopy

Cells were collected from exponential growth, washed twice in acetate buffer (18 mM CH$_3$COONa, 1 mM CaCl$_2$, 1 mM MnCl$_2$, pH 5.2), and immobilized on PDMS stamps as described before in *Formosa et al., 2015a*. AFM experiments were recorded at room temperature using a Nanowizard III system (JPK-Bruker) and MLCT cantilevers (Bruker). The spring constants of each probe were systematically measured by the thermal noise method according to *Jl and Bechhoefer, 1993* and were found to be in the range of 0.01–0.02 N/m. AFM height and adhesion images were recorded in QI mode (JPK-Bruker) (*Chopinet et al., 2013*), and the maximal force applied to the cell was limited to 1 nN. For each condition, three independent experiments were performed and between four and eight cells per replicates were imaged. All results were analysed with the JPK Data Processing software. The adhesion values measured on cells were determined from the retract force-distance curves. The stiffness values ($k_{cell}$) were determined as the slope of the linear portion of the force versus indentation curves according to the equation (*Arnoldi et al., 2000*)

$$k_{cell} = k \left( \frac{s}{1-s} \right)$$

The effective spring constant of the cell $k_{cell}$ is calculated from the experimental slope $s$ of the force curve and the spring constant $k$ of the cantilever measured by the thermal noise method. All stiffness and adhesion values were considered for the histograms, which were generated using OriginPro version 2020 (OriginLab, Northampton, MA). A synthetic peptide based on the amyloid sequence VVSTTV bearing a replacement of V by A (VASTTV) was purchased from Gencust and used as anti-amyloid peptide against Flo11p amyloid sequence. Cells were incubated for 90 min with 2 mg/ml of this peptide prior to AFM analysis.

## Quantitative reverse transcription PCR

Unless otherwise stated, yeast strains were cultured in YPD medium and five DO of exponentially growing cells (OD$_{600 nm}$ ~ 1.5) were harvested by centrifugation, nitrogen-frozen, and stored at –80 °C. RNA extraction was carried out using RNEasy Plus Mini kit (Qiagen) according to manufacturer's protocol. Quality and quantification of RNAs were determined using a Nanodrop 2000 (Thermo) and Bioanalyzer 2100 (Agilent). To synthetize cDNA, 1 µg of RNA was used in a 20 µl reaction mixture using the iScript cDNA synthesis kit (Bio-Rad). Reactions for qRT-PCR were performed in 20 µl reaction with 1 µl cDNA (0.25 ng/µl final concentration), 10 µl of iQ SYBR Green Supermix buffer (Bio-Rad), 5 µl of Nuclease free water, and 4 µl of the appropriate oligonucleotides (*Supplementary file 2*) at a final concentration of 0.25 µM, designed using the qPCR Primer & Probe Design Webtool, and run on a MyIQ real-time PCR system (Bio-Rad). All reactions were run in triplicate, with *TAF10* and *UBC6* used as reference genes (*Teste et al., 2009*). The relative transcript abundances of *FLO11* normalized to TAF10 and UBC6 were calculated based on the 2$^{-\Delta\Delta Ct}$ method (*Livak and Schmittgen, 2001*).

## Immunofluorescence assays

Yeast cells were grown overnight at 30 °C, 200 rpm agitation, in YPD medium. After collection by centrifugation (3000 rpm for 2 min) and washed two times with PBS 1× X, they were then resuspended and fixed into 4 % paraformaldehyde solution for 30 min at room temperature, then washed again three times in PBS 1 ×, and permeabilized in Tween 0.2 % for 15 min at room temperature. Cells were further washed intensively using PBS 1 × and stained in that same buffer using Alexa Fluor 488-conjµgated Anti-6x-His-Tag Monoclonal Antibody (purchased from Thermo Fisher Scientific) at a dilution of 1:25. Cells were left for 1 hr at room temperature in the dark. After staining, cells were gently washed three times in PBS 1×  and mounted on agar on a microscope slide before observation (100×).

## Preparation of cell wall samples for quantification of proteins by shotgun proteomic

Cell wall and membrane fractions from L69 strain expressing Flo11-His and their variants also bearing a His-tag at their C-terminus were prepared using the procedure described by *Sarode et al., 2011* with some modifications. Briefly, the pellet obtained after cell breakage by centrifugation at 10,000 *g* for 30 min was digested 2 hr at 37 °C in 300 µl of a sodium acetate buffer 50 mM pH 5.0 containing a mix of five units endo-β-1,3-glucanase/β-glucosidase (purchased from Megazyme, Irland) and 25 mM of β-mercaptoethanol. This suspension was centrifuged again at 10,000 *g* for 30 min and the pellet was resuspended in 100 µl of Tris HCl buffer 50 mM pH 7.4 containing 2 % DS and boiled for 10 min. Proteins from this last treatment were precipitated with three volumes of cold acetone at 4 °C for 16 hr. Samples were dried in SpeedVac and resuspended in 100 µl water. Before proteomic analysis, an N-deglycosylation was carried out on 200 µg of protein with *x* unit of Endo H at 37 °C for 90 min as recommended by the manufacturer's (New England Biolabs). The proteins were recovered by precipitation with cold acetone as above, were dried, and resuspended in 50 µl of a 5 % SDS solution. Additional protocol for proteomic analysis and listing of identified proteins are reported in *Supplementary file 3*.

## Assay of adherence to polystyrene

The adherence of yeast cells to polystyrene surfaces were carried out according to *Reynolds and Fink, 2001* with some modifications. The cells were grown overnight at 30 °C in YNGal (supplemented with auxotrophic requirements when necessary), washed once in PBS buffer, and resuspended in YNGal adjusted at either pH 5.0 or pH 8.0 with HCL or NaOH 1 M to obtain one unit $OD_{600}$. Aliquots of 100 µl of the culture were transferred into 96-well polystyrene plates and the cell suspensions were incubated at 30 °C for 1 hr at 200 rpm. An equal volume of 0.1 % (w/v) crystal violet was then added to each well. After 15 min, the wells were washed four times with sterile water, and the adherence of the cells was quantified by solubilizing the retained crystal violet in 100 µl 95 % ethanol. After 10 min, the absorbance of the samples against the blank was measured at 595 nm in microplate reader (Biotek).

## Octane adhesion test

Hydrophobic features of yeast surface were determined by measuring their affinity for a nonpolar solvent as described in *Purevdorj-Gage et al., 2007*. Overnight cultures were centrifuged at 2.000 *g* for 5 min and resuspended in fresh YN galactose medium at $OD_{600}$ of 1. After 3 hr of static incubation at room temperature, $OD_{600 \, nm}$ was measured (*A0*) and 1.2 ml of each culture was overlaid with 0.6 ml of octane (Sigma-Aldrich) in 15 × 100 mm borosilicate glass tubes. The tubes were vigorously vortexed for 2 min and left on the bench for at least 15 min until complete separation of the two phases. A sample of the aqueous phase was taken with a Pasteur pipette and the $OD_{600 \, nm}$ was measured (*A*). The results were expressed as the octane adhesion index (% hydrophobicity), which represents the percentage of cells retained by the organic fraction, according to the equation:

$$Hydrophobicity = 1 - \left( \frac{A}{A0} \right) * 100$$

### Invasive agar growth assay

Agar invasion assays were performed according to the method previously described (*Roberts and Fink, 1994*). Strains were patched on glucose or galactose-rich or synthetic medium plates supplemented with auxotrophic compounds when needed and grown at 30 °C for 5–8 days. They were photographed before and after washing with distilled water.

### Cell-cell aggregation assay

Yeast cells were cultivated in a synthetic minimal (YN) medium with 2% galactose (YN Gal) whose initial pH was measured at 5.0. Cell concentration of each sample was adjusted to $8 \cdot 10^7$ cells/ml and 4 µl were dropped on a microscope slide and observed with a microscope Eclipse 400 (Nikon) after brief vortexing. Cell-cell aggregation were imaged using a Zeiss microscope connected to a CMOS camera. Aggregates with five or more cells were counted in three replicated experiments with at least 200 cells scored from three independent microscopic views and reported as percentage of total cells and aggregates.

### Other phenotypic assays

Sensitivity of yeast strains to caffeine (Sigma-Aldrich), CFW (US Bio), and CR was performed on YPD agar plates. Briefly, exponentially growing cells on YPD ($OD_{600 nm}$ around 1) were collected by centrifugation and resuspended in sterilize water at 10 OD units. Series of 10-fold dilutions were spotted on YPD agar plates in the absence or presence of various concentrations of CFW or CR. Pictures were taken after 2 days of growth at 30 °C. Flocculation tests were carried out according to *Lo and Dranginis, 1996*, starting with overnight yeast cultures in YPD, washed once with deflocculating medium (20 mM citrate pH 3.0 containing 5 mM EDTA), and resuspended in 1 ml of the same solution at 1.0 unit $OD_{600}$. Then, CaCl2 (1 M solution) was added at a final concentration of 20 mM and decrease of absorbance was monitored at 600 nm. Velum formation was carried out exactly as described in *Zara et al., 2005*. The flor yeast strain A9 (kind gift from M Budroni, University of Sassari, Italy) was used as a control.

### Statistical analysis

All data of phenotypes assays carried out in this work ware obtained from at least three independent biological experiments. Statistical analysis were made by one-way analysis of variance followed by Tukey's test on Microsoft Excel software. Statistical significant values were denoted by asterisks on the figures as *=p-value < 0.05, **=p-value < 0.01, and ***=p-value < 0.001.

## Acknowledgements

We are grateful Dr Jean Luc Parrou for advice on performing qRT-PCR experiments, to Dr Charlie Boone of University Toronto Canada, and Dr Marilena Budroni from University of Sassari, Italy, for the kind gift of yeast strains, and to Dr Mathieu Castex from Lallemand Inc.Inc for continuous support on this work. CB was financed by an ANRT (Agence Nationale de la Recherche et des Technologies) grant to carry out her PhD thesis. We would also like to thank the imaging and proteomics platforms of Genopole Toulouse (Genotoul at https://www.genotoul.fr/en/) for their valuable technical support in the immunofluorescence experiments of Flo11p and the proteome analysis of the L69 strain. Our thanks also go to Mrs Cathy Botanch of our team for her precious help in the preparation of samples for immunofluorescence analyses.

## Additional information

### Funding

| Funder | Grant reference number | Author |
| --- | --- | --- |
| Region Occitanie | 09003813 | Jean Marie François |
| Lallemand SAS | SAIC2016/048 | Jean Marie François |

| Funder | Grant reference number | Author |
|--------|------------------------|--------|
| Lallemand SAS | SAIC 2018/010 | Jean Marie François |

The funders had no role in study design, data collection and interpretation, or the decision to submit the work for publication.

### Author contributions

Clara Bouyx, Investigation, Methodology, Writing - original draft; Marion Schiavone, Conceptualization, Investigation, Methodology, Writing - original draft; Marie-Ange Teste, Investigation, Methodology; Etienne Dague, Investigation, Methodology, Supervision; Nathalie Sieczkowski, Conceptualization, Methodology; Anne Julien, Conceptualization, Methodology, Supervision; Jean Marie François, Conceptualization, Funding acquisition, Supervision, Writing - original draft, Writing - review and editing

### Author ORCIDs

Marie-Ange Teste (iD) http://orcid.org/0000-0001-9173-9190
Jean Marie François (iD) http://orcid.org/0000-0001-9884-5535

### Decision letter and Author response

Decision letter https://doi.org/10.7554/eLife.68592.sa1
Author response https://doi.org/10.7554/eLife.68592.sa2

## Additional files

### Supplementary files

• Supplementary file 1. TANGO software analysis of β-aggregation motifs in Flo1, Flo5, Flo9, Flo10, and Flo11 protein from *Saccharomyces cerevisiae* S288c strain.

• Supplementary file 2. Identification of sequence repeats and beta-aggregation prone sequence in Flo11 proteins from different yeast strains. (a) Search for intragenic repeats using EMBOSS ETANDEM software. (b) Search for β-aggregation-prone sequence in the different Flo11 proteins using TANGO software (http://tango.crg.es/). *β-Aggregation-prone sequences > 30 % were searched using TANGO software (at http://tango.crg.es/) with default setting of pH, ionic strength, and temperature. Amyloid-core sequences are highlighted in yellow.

• Supplementary file 3. Proteomic analysis of cell wall/cell membrane preparation from L69 strain.

• Supplementary file 4. Yeast strains used or constructed in this study.

• Supplementary file 5. Plasmids constructed in this work.

• Supplementary file 6. Oligonucleotides used in this study.

• Transparent reporting form

### Data availability

The raw dataset has been deposited to Dryad and is accessible at https://doi.org/10/10.5061/dryad.v41ns1rvv. The sequence of the FLO11 gene from the industrial strain used in this study has been deposited at NCBI Gene under the accession number 854836.

The following dataset was generated:

| Author(s) | Year | Dataset title | Dataset URL | Database and Identifier |
|-----------|------|---------------|-------------|-------------------------|
| François JM | 2021 | Data from: Physiological function of Flo11p domains and the particular role of amyloid core sequences of this adhesin in *Saccharomyces cerevisiae* | https://doi.org/10.10.5061/dryad.v41ns1rvv | Dryad Digital Repository, 10.5061/dryad.v41ns1rvv |
| Bouyx C, Schiavone M, Teste MA, Dague E, Sieczkowski N, Julien A, François JM | 2021 | *Saccharomyces cerevisiae* FLO11 (FLO11) gene, complete cds | https://www.ncbi.nlm.nih.gov/gene/854836 | NCBI Gene, 854836 |

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
