## [Decision Letter]

**Acceptance summary:**

Your study confirms and expands our understanding of fungal adhesins in general, and the role of amyloid-forming regions in particular. Specifically, the data show that amyloid sequences are needed for cis-interactions leading to clusters of Flo11p at the cell surface.

**Decision letter after peer review:**

Thank you for submitting your article "Physiological function of Flo11p domains and the particular role of amyloid core sequences of this adhesin in *Saccharomyces cerevisiae*" for consideration by *eLife*. Your article has been reviewed by 3 peer reviewers, and the evaluation has been overseen by a Reviewing Editor and Vivek Malhotra as the Senior Editor. The following individuals involved in review of your submission have agreed to reveal their identity: Peter Lipke (Reviewer #1).

While all reviewers agree that your work is an interesting confirmation of previous findings, they also agreed that the study suffers from several loose ends that need to be resolved before the paper can be considered for publication in *eLife*.

Firstly, there is no direct evidence that the different adhesin constructs are present in more or less equal amounts at the cell surface. Expression levels may not be a good proxy for the levels of active protein at the cell surface, especially for constructs lacking the GPI domain medium and decreased surface anchorage. Hence, further experiments to address this issue, for example using previously established methods to observe adhesins, such as atomic force microscopy and immunofluorescence microscopy, seem essential.

Secondly, the formation of amyloid-like structures was not demonstrated directly. Demonstrating that the relevant peptides can form amyloids that can be broken by ThioS or the breaker peptide, would take care of that issue.

Third, it does not seem appropriate to only show data from one cell instead of reporting the aggregated data from all cells.

A last important issue is that the structure of the abstract and in fact the whole manuscript is perhaps not optimal. The authors convey and mix too many different messages, and the summary and main text would benefit from simplification and a better separation of the different questions that are addressed. In addition, the authors are also encouraged to better discuss their findings against the background of previous literature, explicitly pointing out where their findings confirm previous observations.

*Reviewer #1 (Recommendations for the authors):*

The manuscript would benefit from discussion of relevant data and models in 3 recent papers:

Ho et al., 2019. An Amyloid Core Sequence in the Major *Candida albicans* Adhesin Als1p Mediates Cell-Cell Adhesion. mBio. 2019 Oct 8;10(5):e01766-19. doi: 10.1128/mBio.01766-19.

Dehullu et al., 2019. Fluidic Force Microscopy Captures Amyloid Bonds between Microbial Cells. Trends Microbiol. 2019 Sep;27(9):728-730. doi: 10.1016/j.tim.2019.06.001.

Dehullu et al., 2019. Fluidic Force Microscopy Demonstrates That Homophilic Adhesion by *Candida albicans* Als Proteins Is Mediated by Amyloid Bonds between Cells. Nano Lett. 2019 Jun 12;19(6):3846-3853. doi: 10.1021/acs.nanolett.9b01010.

Specific comments.

l.40: The wall integrity statement is not supported by the data: the data shows that expression of Flo11 contributes to resistance to wall-perturbing drugs. The wording would be modified accordingly

l. 48: "reported that an adhesin…"

l. 48, 58: also shown for Als1p: Ho et al., 2019

l 61 and elsewhere: The authors may also find that Dehullu et al., 2019a, b are useful in data analysis and discussion. These papers describe and summarize formation of amyloid-like bonds between cells to mediate cellular aggregation.

l. 76: It would be helpful to mention that Flo11 is not homologous to Flo1-family adhesins. There is no apparent homology of Flo11 in any region of the Flo1-family proteins.

l. 106-112: the bases for these statements are not convincing

l. 138-141: sentence not clear

l. 159 delete "a" in two places, because the subject is plural

l 157-161: Figure 3A shows ~75% inhibition of the low-adhesion domains as well. This inhibition contradicts the statement.

l. 180: suggest "facilitated" rather than "warranted"

l. 184-187: the sentence contains a double negative "…the lack of nanodomains formation in BY4741 is not solely due…" and would be clearer if rephrased.

l. 206: reference should be Dranginis 2007.

l. 223-225: If the VASTTVT mutant was not made and tested, I suggest alternate wording "…Finally, it is interesting to notice that mutating the VVSTTV to VASTTVT (the sequence used in Figure 2a as anti-amyloid peptide) dropped the TANGO-predicted β-aggregation potential of the in this region of the Flo11 sequence to 26% (Table S3)." Residue position numbers would help.

l. 253: give quantitative estimates or % reduction in stiffness.

l. 254, 321, etc.: All discussion of effects of the δ C-Ter form are suspect because we do not know whether any Flo11 is present on the cell surface, nor how it is attached if present, nor its cell-surface orientation. Labeling and detection of surface Flo11 levels would be required.

l. 262. The adhesion peaks in Figure S10 look like 300-500 pN, rather than the 100 pN stated.

l. 322. The loss of aggregation might also be caused by differences in cell surface concentration.

l. 331: typo: should be "not"

l. 348: differences might also be caused by differences in cell surface concentration due to different culture methods.

l. 418 see comment on l. 40

l. 460-470: This discussion is not clear. The sentence on l. 462-3 appears to be self-contradictory: how can L69 not exhibit a flocculation phenotype? The concluding sentence of the paragraph has a double negative, and its meaning is not clear.

l. 848 has typos: "Figure 6: The Flo11p cell-cell aggregation is potentiated by increased number of amyloid-forming sequences."

Figure 8 overlapping label on 2nd column

Figure S10 correct to '…a number obtained from a single cell of…"

*Reviewer #2 (Recommendations for the authors):*

I find the abstract a bit unfocussed and trying to convey too many messages. Please simplify it.

*Reviewer #3 (Recommendations for the authors):*

In the introduction, include in the discussion the state-of-the art on Flo11p, such the structure Flo11 proteins that have been recently solved as well as functional studies on Flo11p-based cell-cell interaction (including AFM-based single-cell interaction studies),

see

– Brückner S, Schubert R, Kraushaar T, Hartmann R, Hoffmann D, Jelli E, Drescher K, Müller DJ, Oliver Essen L, Mösch HU. Kin discrimination in social yeast is mediated by cell surface receptors of the Flo11 adhesin family. *eLife*. 2020 Apr 14;9:e55587;

– Kraushaar T, Brückner S, Veelders M, Rhinow D, Schreiner F, Birke R, Pagenstecher A, Mösch HU, Essen LO. Interactions by the Fungal Flo11 Adhesin Depend on a Fibronectin Type III-like Adhesin Domain Girdled by Aromatic Bands. Structure. 2015 Jun 2;23(6):1005-17.

Also include recent reviews on fungal adhesins such as

– Willaert RG. Adhesins of Yeasts: Protein Structure and Interactions. J Fungi (Basel). 2018 Oct 27;4(4):119.

– Essen LO, Vogt MS, Mösch HU. Diversity of GPI-anchored fungal adhesins. Biol Chem. 2020 Nov 26;401(12):1389-1405.

Other relevant (landmark) previous research on Flo11p should be included in the introduction and/or the discussion of the results:

– Lambrechts MG, Bauer FF, Marmur J, Pretorius IS. Muc1, a mucin-like protein that is regulated by Mss10, is critical for pseudohyphal differentiation in yeast. Proc Natl Acad Sci U S A. 1996 Aug 6;93(16):8419-24..

– Bayly JC, Douglas LM, Pretorius IS, Bauer FF, Dranginis AM. Characteristics of Flo11-dependent flocculation in *Saccharomyces cerevisiae*. FEMS Yeast Res. 2005 Dec;5(12):1151-6. doi: 10.1016/j.femsyr.2005.05.004.

– Douglas LM, Li L, Yang Y, Dranginis AM. Expression and characterization of the flocculin Flo11/Muc1, a *Saccharomyces cerevisiae* mannoprotein with homotypic properties of adhesion. Eukaryot Cell. 2007 Dec;6(12):2214-21. doi: 10.1128/EC.00284-06.

– Purevdorj-Gage B, Orr ME, Stoodley P, Sheehan KB, Hyman LE. The role of FLO11 in *Saccharomyces cerevisiae* biofilm development in a laboratory-based flow-cell system. FEMS Yeast Res. 2007 May;7(3):372-9.

– Van Mulders SE, Christianen E, Saerens SM, Daenen L, Verbelen PJ, Willaert R, Verstrepen KJ, Delvaux FR. Phenotypic diversity of Flo protein family-mediated adhesion in *Saccharomyces cerevisiae*. FEMS Yeast Res. 2009 Mar;9(2):178-90.

– Goossens KV, Willaert RG. The N-terminal domain of the Flo11 protein from *Saccharomyces cerevisiae* is an adhesin without mannose-binding activity. FEMS Yeast Res. 2012 Feb;12(1):78-87.

p. 4: In the description of the protein architecture the function of the A-domain as an adhesin is now missing and should be described in detail, since it is the A-domain that contains the Flo11 domain (see Pfam database: PF10182 or the interpro database). The Flo11p domain is now generally accepted as the domain of the Flo11 protein that is responsible for the cell-cell interaction.

Nanodomains were observed using AFM and a bare tip. Some details on how the measurements were performed are missing from the Materials and Methods section, such as which cantilever(s) of the MLCT were used, in which buffer (or growth medium) (what was the pH?) were the experiments performed. Were the cantilevers plasma or O3 cleaned?

A question that arises when looking at the adhesion maps and adhesion force determination is which type of interaction is present here? Is it an electrostatic or hydrophobic/hydrophilic interaction? Plasma/O3 cleaning can change the hydrophilicity of the cantilever and the adhesion strength. It would be interesting to better characterize the type of interaction since now there is no control. As an additional measurement, the cantilever could be coated with compounds that make the surface hydrophobic or positive or negative charged.

Additionally, since the type of interaction is not well characterized (as well as the conditions), it is not clear now with which part of the Flo11p (and/or the glycans (Meem 7 Cullen, 2012); since yeast glycans contain (charged) phosphates that can interact (Goossens KV, Ielasi FS, Nookaew I, Stals I, Alonso-Sarduy L, Daenen L, Van Mulders SE, Stassen C, van Eijsden RG, Siewers V, Delvaux FR, Kasas S, Nielsen J, Devreese B, Willaert RG. Molecular mechanism of flocculation self-recognition in yeast and its role in mating and survival. mBio. 2015 Apr 14;6(2):e00427-15). For example, Kraushaar et al., (Structure. 2015 Jun 2;23(6):1005-17) showed that the Flo11 domain trans-interact via a hydrophobic interaction due to the presence of 2 aromatic bands in the structure. They also discovered that this interaction only occurs at low pH (5.5) and not at higher pH (6.8)).

These aspects will influence the results on the measured adhesion forces. A bimodal distribution of the adhesion forces were observed. The weaker adhesion forces were attributed to surface proteins that unfold upon retraction of the AFM tip (p. 6, line 137; p. 7, line 160). Are these surface proteins the Flo11 proteins? There is no direct evidence from the results that these nanodomains are composed of Flo11p. This could be obtained by adhesion maps using an anti-Flo11p antibody or with the purified Flo11p, or Flo11p on other cells by performing single cell force-spectroscopy (Brückner et al., e*Life*. 2020 Apr 14;9:e55587). What was the effect of lowering the maximal applied force on the adhesion forces?

If the weaker adhesion forces were due to unfolding of proteins, this should be visible in the force-distance curves. If it was not visible, this hypothesis should be rejected. Please provide these graphs.

p. 11, line 254: "On the other hand, the cell surface of L69flo11-∆Cter cell was characterized by needle-shaped nanostructures whose height was 3 times greater than nanodomains formed on the surface of the L69 strain. These nanodomains displayed adhesion forces that were scattered from a few pN to max 200 pN and stiffness that was roughly 30% higher than that of nanodomains from L69 strain (see Figure S9, in supplementary data)."

– How could these results be explained? When the C-terminal part of the protein has been removed, is it still anchored in the cell wall? Is there proof of this? Suppose that the truncated Flo11p is present: Does the bare AFM tip interact specifically with the remainder of Flo11p and does cis-interaction results again in these nanodomains? Why are these nanodomains then larger?

The force-distance curves shown in Figure S.9 G, indicate 2 slopes indicating that first a softer layer is pushed through. Not clear what this could be?

p. 12, line 293: …"This residual clumping is likely a specific feature from Σ1278b background strain, which does not hamper the notion that Flo11p is critical in cell-cell interaction. This is further supported by the lack of aggregates in BY4741 as this strain does not express FLO11 due to non-sense mutation in FLO8 encoding its major transcriptional activator.

– Not clear what is meant with a "specific" feature? How could this remaining interaction be explained? Remark that Brückner et al., (2020) demonstrated quantitatively using single-cell force spectroscopy that the Flo11p A-domain is needed to have a significant cell-cell interaction (although some variation is present in these single-cell experiments).

– Were these aggregation experiments also performed at a lower pH and could the results of Kraushaar et al., (2015) be confirmed?

– It has been shown recently that the BY4741 strain can flocculate depending on the condition and this flocculation was independent of Flo5p (see Degreif D, de Rond T, Bertl A, Keasling JD, Budin I. Lipid engineering reveals regulatory roles for membrane fluidity in yeast flocculation and oxygen-limited growth. Metab Eng. 2017 May;41:46-56). This has been shown for Flo1p. In many conditions, Flo1p and Flo11p are be coexpressed. Remark that the expression of FLO11 is very complex and control has been attributed to many transcription factors (Brückner S, Mösch HU. Choosing the right lifestyle: adhesion and development in *Saccharomyces cerevisiae*. FEMS Microbiol Rev. 2012 Jan;36(1):25-58.).

Has it been assessed that the flocs that were observed could be deflocculated by adding EDTA? Was the expression of other Flo proteins also measured?

– How many aggregates were counted to obtain the results? It is not clear if this method can detect significant small changes in interaction forces.

– An additional quantitative method of measuring the interaction between cells is needed to confirm the obtained results based on observing aggregates.

p. 14, line 330: … "Adherence was assayed on polystyrene surfaces."

– Note that this was also investigated before: see Van Mulders et al., 2009, and should be included in the discussion (since here also the adhesion of cells that express other Flo proteins was investigated).

p. 14, line 335: "Moreover, the moderate adherence of BY4741 cell to plastic at pH 5.0 must be due to cell wall proteins other than Flo11p, since this gene is not expressed in this strain due to a non-sense mutation in FLO8 encoding its transcriptional activator (Liu et al., 1996).

– Same comment as before: Under the selected conditions, was it confirmed that no other Flo proteins were expressed?

p. 16 and discussion: Invasive growth experiments => discuss the results also in relation to previous published results on this (such as Van Mulders et al., 2009).

p. 19, line 454: "The finding that this phenotype was not abrogated when domain A was deleted could be at first glance in contradiction with the data of Kraushaar et al., (Kraushaar et al., 2015) who showed that this domain is essential for conferring cell-cell adhesion by facilitating homotypic interactions".

– Also in contrast to others: Bayly et al., (2005), Goossens and Willaert (2012).

However, the lack of A domain can be in part overcome by amyloid-forming sequences as these later could promote clustering of Flo11 molecules in cis, which enables cell-cell adhesion by trans-interaction as proposed by these same authors (Kraushaar et al., 2015, Bruckner et al., 2020).

– In the model hypothesis of Kraushaar et al., 2015; Bruckner et al., 2020 (based and their results as well on previous results from e.g. Ramsook et al.,), they propose that there is trans interaction between interacting cells, which is based on the interaction of the Flo11 domains, and that cis-interaction between various Flo11 proteins via the β-aggregation-prone amyloid-forming sequences could precede the trans-interaction. This model needs further experimental confirmation to be generally accepted.

It could be that the observed nanodomains correspond to trans-interacting Flo11 proteins. However, the cell-cell interaction would be significantly reduced if the Flo11p domains are not present in the truncated protein. This is however in contrast to what has been observed by the authors in this manuscript. Therefore, it is not clear how this can be explained.

---

## [Author Response]

While all reviewers agree that your work is an interesting confirmation of previous findings, they also agreed that the study suffers from several loose ends that need to be resolved before the paper can be considered for publication in eLife.Firstly, there is no direct evidence that the different adhesin constructs are present in more or less equal amounts at the cell surface. Expression levels may not be a good proxy for the levels of active protein at the cell surface, especially for constructs lacking the GPI domain medium and decreased surface anchorage. Hence, further experiments to address this issue, for example using previously established methods to observe adhesins, such as atomic force microscopy and immunofluorescence microscopy, seem essential.

It is indeed clear that there is no direct correlation between expression level of a given gene and level of the encoded protein and so we agreed that qRT-PCR, although showing similar expression levels of the genes encoding the wild type and the variants of Flo11 should be completed by a more direct quantification of the protein. We solve this issue in two ways:

A – As suggested by reviewer #1 and Editor, we carried out immunofluorescence assays. To do this, Flo11 and its variants were flanked at their C-ter of a 6-His tag which was inserted into the gene by CRISPR-cas9. Then, the immunofluorescence assay was carried out using Alexa Fluor 488-conjugated Anti-6x-His Tag Monoclonal Antibody. The result of this experiment clearly shows strong and distinct fluorescence spots at the cell periphery, regardless of the Flo11 variant used. This result, which we present as Figure 5 in the revised paper, leads to two relevant pieces of information which are (i) loss of amyloid β-aggregation sequence does not abrogate this localisation and (ii) more importantly, loss of C-ter and thus of the GPI anchor does not apparently prevent Flo11 protein to be retained at the cell surface. We then wanted to quantify the level of Flo11 protein produced in strains using flow cytometer, but the signal level was close to the autofluorescence, likely because our constructs are expressed at a single copy from the endogenous FLO11 promoter.

B – The second approach was to quantify Flo11p by a shot gun proteomic analysis on samples enriched with cell wall and cell membranes according to a protocol taken from Sarode et al., (EC, 10, 1516-1526, 2011). We did this experiment twice and identified around 700 proteins, from cell wall, cell membrane and mitochondria (likely because of the process of centrifugation that was suggested in the protocol). In cell wall, we identified about 20 cell wall among which Gas1, Fks1, Fks2, cwp1, several cell mannoproteins such as Pir1, Pir4, Pst1, Cwp1, but we failed to detect Flo11p as well as other flocculins. When looking carefully to the median abundance of the cell wall proteins that were identified in this shot-gun proteomic analysis, using SGD database , it turns out that the detection limit for statistical confidence in protein identification roughly corresponded to 3000 molecules per cell (ie the case for EXG2 encoding an endo1,3-glucanase). Thus, it may be possible that level of Flo11p was below this level, which could also explain that we were unable to detect Flo11p by Western blot using the same strain that was used for immunofluoresecence and thus using the anti-HIS antibody to cross react with the 6xHis tag that is at the C-ter of the Flo11p. It should be noticed that detection of Flo11p reported in previous work was done using strains that were transformed with a 2-µ plasmid that carries FLO11 under a strong promoter such as PGK1 (see work of Douglas et al., 2007) or under GAL1. Thus, the data we have reported here actually suggest that Flo11p is not a very abundant protein and that if this protein is highly overproduced, this may result in the expression of phenotypic behaviours that no longer reflect physiological reality. For the interest of reviewers and readers, we have provided the proteomic data for strain L69 in the supplementary data (data of one typical proteomic analysis is reported Supplementary File 3, as the second provided very similar results).

Secondly, the formation of amyloid-like structures was not demonstrated directly. Demonstrating that the relevant peptides can form amyloids that can be broken by ThioS or the breaker peptide, would take care of that issue.

It was already shown that Flo11p assembled into amyloid fibers in vitro (see reference Ramsook et al., Eukaryotic Cell, 9: 393-404, and this reference is given in our paper). This was not the purpose of this work to rediscover that, but to show that the nanodomains at the surface of the L69 strain was due to the presence of amyloid-core sequences in the Flo11p. Our data support the Lipke’ model that these β-aggregation prone sequences are required for the production of these nanostructures in response to an external force such as the retraction of AFM tip from the surface, and that these nanostructures are abolished when these sequences are removed in the protein (Flo11-ΔRR2) or upon incubation with an anti-amyloid peptide or amyloid perturbing agents such as thioflavin S.

Third, it does not seem appropriate to only show data from one cell instead of reporting the aggregated data from all cells.

As indicated in MandM, all AFM experiments were repeated 3 times on each strain expressing a Flo11 variant (ie 3 biological replicates), with 4 to 8 cells per replicate. For each cell, 1024 force-distance curves have been recorded given rise to histograms of the distribution of adhesion forces or spring constants. In figure 1E, we reported such a representation for one representative cell. By aggregating all data, we found obviously a similar bimodal distribution of adhesion values, characterized however by a high variability. Thus, we suggested to present the aggregated values in the manuscript as box plots (see figure 1 Gand H) to illustrate both this bimodality of the distribution and the variability of the adhesion force and stiffness.

A last important issue is that the structure of the abstract and in fact the whole manuscript is perhaps not optimal. The authors convey and mix too many different messages, and the summary and main text would benefit from simplification and a better separation of the different questions that are addressed. In addition, the authors are also encouraged to better discuss their findings against the background of previous literature, explicitly pointing out where their findings confirm previous observations.

We have extensively revised our paper to focus on the importance and role of amyloid sequences in the function of Flo11 in yeast and in particular, our data show that amyloid sequences are needed for cis-interaction leading to clusters of Flo11p making nanodomain. Also they may contribute to trans-interaction since ablation of these sequences strongly reduce cell-cell interaction, or alternatively, insertion of the RR2 domain from Flo11 of L69 strain that is enriched in amyloid core sequences in the Flo11 of BY strain leads to strong cell-cell aggregation, what barely occurred in the BY strain expressing its normal Flo11p. We thus suggested this new title to "The dual role of the amyloid core sequences in the physiological function of FLO11-encoded flocculin in the yeast *Saccharomyces cerevisiae*".

Reviewer #1 (Recommendations for the authors):The manuscript would benefit from discussion of relevant data and models in 3 recent papers:Ho et al., 2019. An Amyloid Core Sequence in the Major Candida albicans Adhesin Als1p Mediates Cell-Cell Adhesion. mBio. 2019 Oct 8;10(5):e01766-19. doi: 10.1128/mBio.01766-19.Dehullu et al., 2019. Fluidic Force Microscopy Captures Amyloid Bonds between Microbial Cells. Trends Microbiol. 2019 Sep;27(9):728-730. doi: 10.1016/j.tim.2019.06.001.Dehullu et al., 2019. Fluidic Force Microscopy Demonstrates That Homophilic Adhesion by Candida albicans Als Proteins Is Mediated by Amyloid Bonds between Cells. Nano Lett. 2019 Jun 12;19(6):3846-3853. doi: 10.1021/acs.nanolett.9b01010.

We are very grateful to this reviewer for alerting us to these 3 papers. They are in fact extremely relevant for our paper because they consolidate our experimental data suggesting that the amyloid core sequences are needed for nanoclustering of Flo11 molecules by cis-interaction, but also can contribute to cell-cell adhesion by trans-interaction or, alternatively, that this homophilic adhesion which is dependent on the A-domain of Flo11p according to the work of Kraushaar et al., (Structure, 2015) may require amyloid-β sheet interaction sequence to be the most effective. This suggestion is based on our data that cell-cell aggregation are strongly diminished in a strain expressing a Flo11 variant lacking amyloid core sequences, whereas this phenotype is remarkably exacerbated in BY cells either upon expression of the FLO11 of L69 strain or insertion of additional amyloid-core sequences in its endogenous FLO11.

Specific comments.l.40: The wall integrity statement is not supported by the data: the data shows that expression of Flo11 contributes to resistance to wall-perturbing drugs. The wording would be modified accordingly

We showed that loss of Flo11 results in a higher sensitivity to cell wall drugs, which are currently used to assess the cell wall integrity. We do not mean that Flo11/flocculin is needed for wall integrity but only suggested that it may contribute to it. We reformulated therefore this meaning in the revised version (see line 413-415).

l. 48: "reported that an adhesin…"

corrected

l. 48, 58: also shown for Als1p: Ho et al., 2019

corrected

l 61 and elsewhere: The authors may also find that Dehullu et al., 2019a, b are useful in data analysis and discussion. These papers describe and summarize formation of amyloid-like bonds between cells to mediate cellular aggregation.

Indeed, this is a good suggestion and we have included these excellent papers in our manuscript (see notably Introduction).

l. 76: It would be helpful to mention that Flo11 is not homologous to Flo1-family adhesins. There is no apparent homology of Flo11 in any region of the Flo1-family proteins.

Indeed, there are not homologous but we stated about the fact that these flocculins harbor a similar sequence architecture divided in three major region A, B and C.

l. 106-112: the bases for these statements are not convincing

Is it a question of formulation? In any case, we have revised this part of the introduction and written that:

This has been revised (-see line 95-105)

l. 138-141: sentence not clear

This has been revised (see now line 131 -135)

l. 159 delete "a" in two places, because the subject is plural

Corrected

l 157-161: Figure 3A shows ~75% inhibition of the low-adhesion domains as well. This inhibition contradicts the statement.

This sentence and this part of the results have been revised; See line (140 to 151 in the revised version).

l. 180: suggest "facilitated" rather than "warranted"

Corrected

l. 184-187: the sentence contains a double negative "…the lack of nanodomains formation in BY4741 is not solely due…" and would be clearer if rephrased.

Sentence has been rephrased (line 159 -170).

l. 206: reference should be Dranginis 2007.

Indeed, this is corrected.

l. 223-225: If the VASTTVT mutant was not made and tested, I suggest alternate wording "…Finally, it is interesting to notice that mutating the VVSTTV to VASTTVT (the sequence used in Figure 2a as anti-amyloid peptide) dropped the TANGO-predicted β-aggregation potential of the in this region of the Flo11 sequence to 26% (Table S3)." Residue position numbers would help.

Indeed, the VASTTVT variant was not made and tested experimentally. This was purely a value obtained by simulation using TANGO. Upon revision of this paper, it came to us that this sentence should be removed as it did not provide any useful information.

l. 253: give quantitative estimates or % reduction in stiffness.

We explicitly wrote that the stiffness of nanodomains on the cell surface of the Flo11-Nter expressing strain was 60% lower than that of the L69 strain.

l. 254, 321, etc.: All discussion of effects of the δ C-Ter form are suspect because we do not know whether any Flo11 is present on the cell surface, nor how it is attached if present, nor its cell-surface orientation. Labeling and detection of surface Flo11 levels would be required.

See comments above in reply to the Editor who also considered that this comment was very crucial to the paper. Overall, our immunofluorescence assay showed that all Flo11 variants include the one lacking the C-ter which bears the GPI anchor are clearly localized at the cell periphery like the normal protein.

l. 262. The adhesion peaks in Figure S10 look like 300-500 pN, rather than the 100 pN stated.

You are right indeed, but it must be insisted that the presence of these patches were extremely difficult to track on the cell surface, as they appeared only on few cells other did not show that peak. So, we did not want therefore to expand more on these peak but just mention that if they exist, they are really different from nanodomains formed by normal Flo11p.

l. 322. The loss of aggregation might also be caused by differences in cell surface concentration.

This is also a possibility although our qRT-PCR combined with the immunofluorescence may suggest that the cell surface concentration of this Flo11 construct is comparable to that in the wild type strain.

l. 331: typo: should be "not"

Corrected

l. 348: differences might also be caused by differences in cell surface concentration due to different culture methods.

This is indeed a possible reason that cannot be excluded. In addition, we found that the pH effect was not comparable between strains, which could be due to difference in expression of Flo11p with respect to pH, and not the vulture media in our condition because all strains were cultivated in the same condition. We however did not investigate further this question.

l. 418 see comment on l. 40

We do not grasp this point since the sentence only says that the strain expressing Flo11 defective in C-ter is more sensitive to cell wall drugs. We reformulate this sentence (see line 412-415).

l. 460-470: This discussion is not clear. The sentence on l. 462-3 appears to be self-contradictory: how can L69 not exhibit a flocculation phenotype? The concluding sentence of the paragraph has a double negative, and its meaning is not clear.

We completely revised this part of the discussion (now see lines 557 – 566), in particular to take into account this very justified comment.

l. 848 has typos: "Figure 6: The Flo11p cell-cell aggregation is potentiated by increased number of amyloid-forming sequences."

Corrected

Figure 8 overlapping label on 2nd column

Corrected

Figure S10 correct to '…a number obtained from a single cell of…"

The legend has been corrected.

Reviewer #2 (Recommendations for the authors):I find the abstract a bit unfocussed and trying to convey too many messages. Please simplify it.

We have fully revised the abstract taking into account the various comments raised by the reviewers and editors and we now hope it is more in focus with the content of the paper.

Reviewer #3 (Recommendations for the authors):In the introduction, include in the discussion the state-of-the art on Flo11p, such the structure Flo11 proteins that have been recently solved as well as functional studies on Flo11p-based cell-cell interaction (including AFM-based single-cell interaction studies),see– Brückner S, Schubert R, Kraushaar T, Hartmann R, Hoffmann D, Jelli E, Drescher K, Müller DJ, Oliver Essen L, Mösch HU. Kin discrimination in social yeast is mediated by cell surface receptors of the Flo11 adhesin family. eLife. 2020 Apr 14;9:e55587;– Kraushaar T, Brückner S, Veelders M, Rhinow D, Schreiner F, Birke R, Pagenstecher A, Mösch HU, Essen LO. Interactions by the Fungal Flo11 Adhesin Depend on a Fibronectin Type III-like Adhesin Domain Girdled by Aromatic Bands. Structure. 2015 Jun 2;23(6):1005-17.Also include recent reviews on fungal adhesins such as– Willaert RG. Adhesins of Yeasts: Protein Structure and Interactions. J Fungi (Basel). 2018 Oct 27;4(4):119.– Essen LO, Vogt MS, Mösch HU. Diversity of GPI-anchored fungal adhesins. Biol Chem. 2020 Nov 26;401(12):1389-1405.Other relevant (landmark) previous research on Flo11p should be included in the introduction and/or the discussion of the results:– Lambrechts MG, Bauer FF, Marmur J, Pretorius IS. Muc1, a mucin-like protein that is regulated by Mss10, is critical for pseudohyphal differentiation in yeast. Proc Natl Acad Sci U S A. 1996 Aug 6;93(16):8419-24..– Bayly JC, Douglas LM, Pretorius IS, Bauer FF, Dranginis AM. Characteristics of Flo11-dependent flocculation in *Saccharomyces cerevisiae*. FEMS Yeast Res. 2005 Dec;5(12):1151-6. doi: 10.1016/j.femsyr.2005.05.004.– Douglas LM, Li L, Yang Y, Dranginis AM. Expression and characterization of the flocculin Flo11/Muc1, a *Saccharomyces cerevisiae* mannoprotein with homotypic properties of adhesion. Eukaryot Cell. 2007 Dec;6(12):2214-21. doi: 10.1128/EC.00284-06.– Purevdorj-Gage B, Orr ME, Stoodley P, Sheehan KB, Hyman LE. The role of FLO11 in *Saccharomyces cerevisiae* biofilm development in a laboratory-based flow-cell system. FEMS Yeast Res. 2007 May;7(3):372-9.– Van Mulders SE, Christianen E, Saerens SM, Daenen L, Verbelen PJ, Willaert R, Verstrepen KJ, Delvaux FR. Phenotypic diversity of Flo protein family-mediated adhesion in *Saccharomyces cerevisiae*. FEMS Yeast Res. 2009 Mar;9(2):178-90.– Goossens KV, Willaert RG. The N-terminal domain of the Flo11 protein from *Saccharomyces cerevisiae* is an adhesin without mannose-binding activity. FEMS Yeast Res. 2012 Feb;12(1):78-87.

All these papers are indeed relevant for the paper. We have revised the Introduction (see ie lines 78 to 90 in the revised version), taking into account some of the data related to Flo11 and data in these paper are more extensively considered in the Discussion section (and most of them have been mentioned in the initial submission) with regards to our data.

p. 4: In the description of the protein architecture the function of the A-domain as an adhesin is now missing and should be described in detail, since it is the A-domain that contains the Flo11 domain (see Pfam database: PF10182 or the interpro database). The Flo11p domain is now generally accepted as the domain of the Flo11 protein that is responsible for the cell-cell interaction.

We have provided some additional description related to A-domain in the revised manuscript (see line 68 to 84), but as written above, discussed these important data in regards to our data (see line 484 -514).

Nanodomains were observed using AFM and a bare tip. Some details on how the measurements were performed are missing from the Materials and Methods section, such as which cantilever(s) of the MLCT were used, in which buffer (or growth medium) (what was the pH?) were the experiments performed. Were the cantilevers plasma or O3 cleaned?

All these technical details are given in the Mand M, except that we did not specify that plasma with air was used to clean the cantilever before the experiment which has now been added.

A question that arises when looking at the adhesion maps and adhesion force determination is which type of interaction is present here? Is it an electrostatic or hydrophobic/hydrophilic interaction? Plasma/O3 cleaning can change the hydrophilicity of the cantilever and the adhesion strength. It would be interesting to better characterize the type of interaction since now there is no control. As an additional measurement, the cantilever could be coated with compounds that make the surface hydrophobic or positive or negative charged.Additionally, since the type of interaction is not well characterized (as well as the conditions), it is not clear now with which part of the Flo11p (and/or the glycans (Meem 7 Cullen, 2012); since yeast glycans contain (charged) phosphates that can interact (Goossens KV, Ielasi FS, Nookaew I, Stals I, Alonso-Sarduy L, Daenen L, Van Mulders SE, Stassen C, van Eijsden RG, Siewers V, Delvaux FR, Kasas S, Nielsen J, Devreese B, Willaert RG. Molecular mechanism of flocculation self-recognition in yeast and its role in mating and survival. mBio. 2015 Apr 14;6(2):e00427-15). For example, Kraushaar et al., (Structure. 2015 Jun 2;23(6):1005-17) showed that the Flo11 domain trans-interact via a hydrophobic interaction due to the presence of 2 aromatic bands in the structure. They also discovered that this interaction only occurs at low pH (5.5) and not at higher pH (6.8)).These aspects will influence the results on the measured adhesion forces. A bimodal distribution of the adhesion forces were observed. The weaker adhesion forces were attributed to surface proteins that unfold upon retraction of the AFM tip (p. 6, line 137; p. 7, line 160). Are these surface proteins the Flo11 proteins?

We do not fully understand your comment and do not see if there is a question behind. What can be commented on our side are:

A – AFM techniques used is all what is the more common.

B – The adhesion forces (what you mention as interaction) have been analysed in detail for each strain expressing the different variant. Only nanodomains from L69 strain expressing the wild type Flo11 exhibits a bimodal distribution of adhesion forces.

C – The weaker adhesion force is likely due to protein being unfolded, which can be Flo11 but not exclusively indeed as we used a bare tip that interacts with cell surface component in general, not specifically. This can be seen in Figure 3 in which this low-adhesion forces was still observed but a lower frequency upon treatment of the cells with the anti-amyloid peptide.

There is no direct evidence from the results that these nanodomains are composed of Flo11p. This could be obtained by adhesion maps using an anti-Flo11p antibody or with the purified Flo11p, or Flo11p on other cells by performing single cell force-spectroscopy (Brückner et al., eLife. 2020 Apr 14;9:e55587).

The evidence that the nanodomains are composed of Flo11p are:

A – deletion of FLO11 in strain L69 completely abrogates the formation of adhesive nanodomains upon AFM tip.

B – the laboratory BY strain that does not form nanodomains with its own FLO11 gene even after overexpression, but only upon ectopic expression of this peculiar Flo11p from strain L69.

C – nanodomains are not produced in cells that express a Flo11 lacking its amyloid-core –sequence.

D – the finding that purified Flo11p forms amyloids in vitro has been already reported by Lipke group (Ramsook, EC, 2010). This is the most direct evidence that nanodomains are made from clusters of Flo11p!

What was the effect of lowering the maximal applied force on the adhesion forces?

We did not carry out this experiment and do not believe it will bring anything new.

If the weaker adhesion forces were due to unfolding of proteins, this should be visible in the force-distance curves. If it was not visible, this hypothesis should be rejected. Please provide these graphs.

This is indeed visible, as indicated in Figure 2E or Figure 3A, Figure 5 —figure supplement 1 and 2 an example. All raw data have been deposited at https://datadryad.org/

p. 11, line 254: "On the other hand, the cell surface of L69flo11-∆Cter cell was characterized by needle-shaped nanostructures whose height was 3 times greater than nanodomains formed on the surface of the L69 strain. These nanodomains displayed adhesion forces that were scattered from a few pN to max 200 pN and stiffness that was roughly 30% higher than that of nanodomains from L69 strain (see Figure S9, in supplementary data)."– How could these results be explained? When the C-terminal part of the protein has been removed, is it still anchored in the cell wall? Is there proof of this? Suppose that the truncated Flo11p is present: Does the bare AFM tip interact specifically with the remainder of Flo11p and does cis-interaction results again in these nanodomains? Why are these nanodomains then larger?The force-distance curves shown in Figure S.9 G, indicate 2 slopes indicating that first a softer layer is pushed through. Not clear what this could be?

As reported, we carried out immunofluorescence assay with all the Flo11 variants and reported in a new figure (Figure 4) that they have the same localization as the wild type (so clearly marked at the cell periphery), indicating that the loss of C-ter does not prevent anchorage of Flo11 at the cell wall. How this protein could be retained would need additional work that is out of the scope of this study.

We then performed AFM analysis on cells expressing these Flo11 variants and with the exception of Flo11-RR2, they all showed nanodomains formation with however a very different typology and a loss of bimodal behaviour. There is no reason that the tip does not interact with the reminder of the Flo11 protein, but we cannot exclude that the expression of this truncated form of Flo11p has somehow altered the cell surface, leading to this adhesion histogram. However, these data indicated that the high adhesion forces that result from amyloid-dependent clustering of Flo11p are no longer present. We have carefully revised our paper to take into account this remark (see now lines 266 -282 in the revised version).

Finally, the force-distance curves shown in Figure 9G is an example for the specific cell of this figure. It shows 2 slopes, but the stiffness has been calculated on the first one.

p. 12, line 293: …"This residual clumping is likely a specific feature from Σ1278b background strain, which does not hamper the notion that Flo11p is critical in cell-cell interaction. This is further supported by the lack of aggregates in BY4741 as this strain does not express FLO11 due to non-sense mutation in FLO8 encoding its major transcriptional activator.– Not clear what is meant with a "specific" feature? How could this remaining interaction be explained? Remark that Brückner et al., (2020) demonstrated quantitatively using single-cell force spectroscopy that the Flo11p A-domain is needed to have a significant cell-cell interaction (although some variation is present in these single-cell experiments).– Were these aggregation experiments also performed at a lower pH and could the results of Kraushaar et al., (2015) be confirmed?

The ∑1278 haploid strain exhibits a very different aggregation phenotype with respect to L29 strain, as deletion of FLO11 did not completely abolish this phenotype as reported in Figure 6 (now 7 in the revised version). Whether the remaining aggregation is due to the expression of another FLO gene is unlikely since Flo11p is the only flocculin expressed in this strain. As FLO gene is under silencing, it cannot be excluded that deletion of FLO11 has allowed desilencing of some FLO through epigenetic mechanism as (reported by Halme et al., Cell, 116, 405-415, 2004). This is only a speculation because we did not verify this suggestion. We revised this paragraph accordingly (see lines 302-312).

Our experiments of cell-cell interaction in YNGal agar was made at pH 5.0 which is a condition reported by Kraushaar et al., to favour hemophilic Flo11-Flo11 interaction. Our results are in part in support the finding of Kraushaar et al., although we did not show it directly. Although our data show that amyloid-core sequence contribute to cell-cell interaction (see figure 6), they are not absolutely essential for this phenotype as shown with the lack of effect of the anti-amyloid agents thioflavin S to be unable to disrupt aggregates formed with cells from ∑1278b strain. This data indirectly indicates that other components are implied in this phenotype, which is likely due to the A-domain (revised in lines 312 -315).

Finally, we tested the adherence at different pH and confirmed that this phenotype was sensitive to pH (hence higher at acidic than alkali pH, see figure 7). But surprisingly, we found that this pH effect was not similar between yeast strains. While strains ∑1278C showed better adherence at pH 8.0 than pH 5.0, this was the opposite for L69 strain and BY4741!. This difference would need additional experiments that is not in the scope of this work.

In our hands, the contribution of the N-terminal domain was more important at pH 8.0 than pH 5.0 for the Flo11p of L29 strain, whereas the amyloid core sequence has no role. However, what was striking was the finding that the adherence to plastic in quantitative term was dramatically increases upon overexpression of FLO11 gene whatsoever the origin of the gene (either FLO11 from BY or from L69). This result somehow alerted us because in the experiments carried out by Kraushaar et al., (Structure, 2016), FLO11 is expressed in CEN plasmid under the dependence of the strong PGK1 promoter. Thus, it may be possible that under these conditions, the effect of removal A-domain was quantitatively more effective.

– It has been shown recently that the BY4741 strain can flocculate depending on the condition and this flocculation was independent of Flo5p (see Degreif D, de Rond T, Bertl A, Keasling JD, Budin I. Lipid engineering reveals regulatory roles for membrane fluidity in yeast flocculation and oxygen-limited growth. Metab Eng. 2017 May;41:46-56). This has been shown for Flo1p. In many conditions, Flo1p and Flo11p are be coexpressed. Remark that the expression of FLO11 is very complex and control has been attributed to many transcription factors (Brückner S, Mösch HU. Choosing the right lifestyle: adhesion and development in *Saccharomyces cerevisiae*. FEMS Microbiol Rev. 2012 Jan;36(1):25-58.).Has it been assessed that the flocs that were observed could be deflocculated by adding EDTA? Was the expression of other Flo proteins also measured?– How many aggregates were counted to obtain the results? It is not clear if this method can detect significant small changes in interaction forces.– An additional quantitative method of measuring the interaction between cells is needed to confirm the obtained results based on observing aggregates.

BY strain bears a non-sense mutation in FLO8 which prevents expression of FLO1 and FLO11 (Liu et al., Genetics, 144, 967-978, 1996; Fichtner et al., Mol Microbiol., 66: 1276-1289, 2007). Van Mulders et al., (FEMS Yeast Res, 9: 178 -190, 2009) did an extensive functional analysis of expressing each of the FLO gene in this strain background, and get data related the phenotype of each FLO genes, which are in line with our data.

In a previous paper (Schiavone et al., FEMS Yeast Res 2015), we reported transcriptomic data on L69 strain showed that FLO11 was the most expressed, but we also identified very weak expression of FLO1, FLO5 and FLO10, as well as FIGURE 2.

The procedure for counting cell aggregates are described more extensively now in M and M SCFS has been used by Bruckner et al., (eLife 2020) and Fluid Force microscopy as developed by Dufrene and coll (see Nanoletters, 2019) could be a good mean to measure the force between cells. We believe that this is not needed for the present work but these experiments are considered in a future work.

p. 14, line 330: … "Adherence was assayed on polystyrene surfaces."– Note that this was also investigated before: see Van Mulders et al., 2009, and should be included in the discussion (since here also the adhesion of cells that express other Flo proteins was investigated).

This paper has been referenced but the methodology was not established by these authors but by Reynolds and Fink (Science 2011). Also, we only restricted the discussion to Flo11 protein and its variant an not to the other Flo protein which was not relevant in this paper.

p. 14, line 335: "Moreover, the moderate adherence of BY4741 cell to plastic at pH 5.0 must be due to cell wall proteins other than Flo11p, since this gene is not expressed in this strain due to a non-sense mutation in FLO8 encoding its transcriptional activator (Liu et al., 1996).– Same comment as before: Under the selected conditions, was it confirmed that no other Flo proteins were expressed?

We did not check for expression of other FLO gene in BY strain. But we may refer to the paper of Halme et al., (Cell, 116, 405-415, 2004) that FLO gene is under epigenetic regulation, and that their expression is metastably silenced. Although this regulation was examined in ∑1272b strain, it might be also valuable in BY strain. We have rephrased our discussion taking into account these data, although this is quite speculative and this suggestion shall merit to be verified, which is out of the scope of this work (see lines 290 -294).

p. 16 and discussion: Invasive growth experiments => discuss the results also in relation to previous published results on this (such as Van Mulders et al., 2009).

Here, we compared the invasion phenotype between Flo11 variant not between different Flo proteins, thus except indicating that our invasive phenotype of FLO11 from BY strain agrees with data of Van Muders et al., there is no need to expend much on their results.

p. 19, line 454: "The finding that this phenotype was not abrogated when domain A was deleted could be at first glance in contradiction with the data of Kraushaar et al., (Kraushaar et al., 2015) who showed that this domain is essential for conferring cell-cell adhesion by facilitating homotypic interactions".– Also in contrast to others: Bayly et al., (2005), Goossens and Willaert (2012).However, the lack of A domain can be in part overcome by amyloid-forming sequences as these later could promote clustering of Flo11 molecules in cis, which enables cell-cell adhesion by trans-interaction as proposed by these same authors (Kraushaar et al., 2015, Bruckner et al., 2020).– In the model hypothesis of Kraushaar et al., 2015; Bruckner et al., 2020 (based and their results as well on previous results from e.g. Ramsook et al.,), they propose that there is trans interaction between interacting cells, which is based on the interaction of the Flo11 domains, and that cis-interaction between various Flo11 proteins via the β-aggregation-prone amyloid-forming sequences could precede the trans-interaction. This model needs further experimental confirmation to be generally accepted.

Recent data from Lipke et coll (see Ho et al., mBIO. mBio. 2019 Oct 8;10(5):e01766-19. doi: 10.1128/mBio.01766-19; Dehullu et al., 2019, Nano Lett. 2019 Jun 12;19(6):3846-3853. doi: 10.1021/acs.nanolett.9b01010) have now clearly demonstrated that amyloid-forming sequences are required for cis-interaction (interaction between adhesion molecule) leading to cluster of adhesion molecule and to trans-interaction, ie interaction cell to cell. Although this demonstration was done with ALS1 and ALS5 encoding Als adhesion of C. albicans, our data are in line with this statement. Indeed, we show that the nanodomains at the cell surface of L69 strain are due to the presence of several amyloid forming sequences present in this particular Flo11p, and this leads to cis-interaction which can be abolished using either the anti-amyloid peptide or an amyloid perturbant, and likely to trans-interaction because loss of RR2 domain that contains these amyloid sequences results in a strong reduction of cell-cell interaction (or cell-cell aggregation).

It could be that the observed nanodomains correspond to trans-interacting Flo11 proteins. However, the cell-cell interaction would be significantly reduced if the Flo11p domains are not present in the truncated protein. This is however in contrast to what has been observed by the authors in this manuscript. Therefore, it is not clear how this can be explained.

This is not what we show in Figure 6 and 7 since the loss of RR2 that contains the amyloid forming sequences results in > 70% reduction of cell-cell interaction, whereas insertion of this fragment in the Flo11 from BY strain allows cells of this strain to form aggregate at a rate that higher than by ectopic overexpression of its endogenous FLO11 (see figure 6).